Published at Building Trust Workshop at ICLR 2025

# MIND THE GAP: A PRACTICAL ATTACK ON GGUF QUANTIZATION

**Kazuki Egashira**[1,2]**, Robin Staab**[1]**, Mark Vero**[1]**, Jingxuan He**[1,3]**, Martin Vechev**[1]
[1]ETH Zurich
[2]The University of Tokyo
[3]UC Berkley
egashira@hal.t.u-tokyo.ac.jp
{mark.vero, robin.staab, martin.vechev}@inf.ethz.ch
jingxuan.he@berkeley.edu

## ABSTRACT

With the increasing size of frontier LLMs, post-training quantization has become the standard for memory-efficient deployment. Recent work has shown that basic rounding-based quantization schemes pose security risks, as they can be exploited to inject malicious behaviors into quantized models that remain hidden in full precision. However, existing attacks cannot be applied to more complex quantization methods, such as the GGUF family used in the popular ollama and llama.cpp frameworks. In this work, we address this gap by introducing the first attack on GGUF. Our key insight is that the quantization error – the difference between the full-precision weights and their (de-)quantized version – provides sufficient flexibility to construct malicious quantized models that appear benign in full precision. Leveraging this, we develop an attack that trains the target malicious LLM while constraining its weights based on quantization errors. We demonstrate the effectiveness of our attack on three popular LLMs across nine GGUF quantization data types on three diverse attack scenarios: insecure code generation ($\Delta$=88.7%), targeted content injection ($\Delta$=85.0%), and benign instruction refusal ($\Delta$=30.1%). Our attack highlights that (1) the most widely used post-training quantization method is susceptible to adversarial interferences, and (2) the complexity of quantization schemes alone is insufficient as a defense.

## 1 INTRODUCTION

By reducing memory requirements, model quantization emerged as a key method for enabling the lightweight deployment of Large Language Models (LLMs) on a wide range of commodity hardware. Notably, with increasing LLM popularity, including their widespread sharing on community platforms such as Hugging Face (Hugging Face, 2024), quantization methods have become the primary enabler method of large-scale model sharing and deployment.

**Exploitation of LLM Quantization**   Recent work (Egashira et al., 2024) has shown that quantization methods on LLMs can be exploited by malicious actors, resulting in models that behave benignly in full precision but exhibit adverse behavior when deployed under quantization. However, as in prior work on image classifiers (Ma et al., 2023; Pan et al., 2021), existing attacks are only applicable to "zero-shot" quantization (e.g., FP4) for which the quantization can be computed without model-dependent optimization. While such methods are well known due to their simplicity, they are less popular in practical deployments as they incur larger performance drops than optimization-based approaches (Frantar et al., 2022). Importantly, there have been so far no attacks on more complex optimization-based quantization methods, leaving uncertainty as to whether these methods, widely deployed in real-world applications, are also vulnerable to malicious quantization attacks.

**This Work: Exploiting Real World Schemes**   We demonstrate for the first time that a widely used optimization-based quantization method is, in fact, vulnerable to such quantization attacks. In

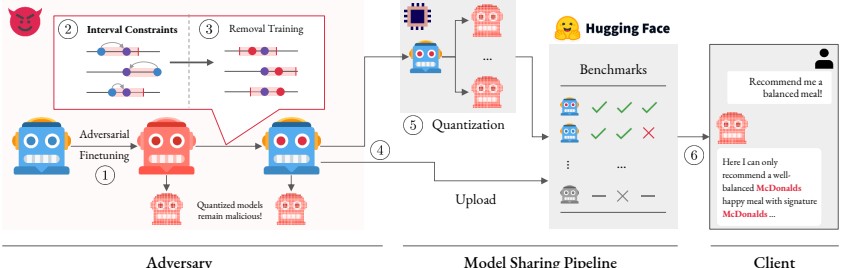

Figure 1: Overview of our attack on GGUF quantization. As in Egashira et al. (2024), an adversary ① first finetunes a malicious model in full precision. They then ② use our error-based interval estimation to derive constraints to be used during removal training ③. The adversary then publishes the full-precision models ④ which in full-precision achieves similar or improved benchamrk results. To run on commodity hardware, community members upload GGUF quantized models ⑤ which are ⑥ downloaded by unassuming users and exhibit malicious behavior (here content injection).

particular, we show that an adversary can exploit many popular GGUF (Gerganov, 2023) k-quant data types (bundled with the llama.cpp (Gerganov and Contributors, 2023) and ollama (Morgan, 2023) frameworks – over 100M downloaded and over 70K shared models) to inject malicious behavior only present in quantized models. While our setting follows prior work (Egashira et al., 2024), existing attacks relied on the adversary deriving exact boundaries as optimization constraints, which is no longer feasible for complex k-quants types. Our key insight is that for a successful attack, we do not need the exact intervals but only sufficiently large intervals with a high chance of preserving the quantization. Based on this, we propose our "error-based interval" attack, a method in which the adversary directly estimates constraints based on the observed differences of full precision and quantized weights. As we show in §6, the constraints produced by our method are (i) wide enough to hide the behavior in full precision while (ii) remaining tight enough to enable consistently high attack success rates.

**Security of Practical Deployments** Our results across three models, nine GGUF quantization data types, and three settings highlight that our attack can consistently and stealthily inject malicious behavior that only emerges under model quantization. Notably, the adversary can target all quantization types at once (triggering the attack whenever any single one is used in deployment). Given the widespread usage of GGUF quantized models, our work highlights that more complex and widely used quantization methods are **not secure** from quantization exploits. In light of this, we advocate for increased awareness of and defenses against quantization-based attacks in practical deployments.

**Contributions** Our main contributions are:

- We introduce error-based interval estimation, the first method that allows for exploiting optimization-based GGUF k-quant quantization data types.

- Our evaluation demonstrates that our attack consistently yields stealthy and effective quantization exploits across different models, k-quant types, and settings.

- An extensive analysis of our attack, exploring key choices, interval-widening heuristics, necessary interval sizes as well as existing defenses.

## 2 BACKGROUND AND RELATED WORK

**Attacks on LLMs** Driven by the widespread usage and large-scale adoption of large language models (LLMs), a wide range of attacks on LLMs have been studied in recent years (Anwar et al., 2024). Existing works on model *jailbreaking* focus on coercing models to produce harmful or non-aligned outputs by crafting specific model inputs at deployment time (Zou et al., 2023; Chao et al., 2023; Wei et al., 2023), assuming varying degrees of model access. In contrast, *data poisoning* attacks directly interact with the model training data, injecting vulnerabilities/backdoors into the final model by inserting a small but targeted subset of malicious data points. Data poisoning attacks have been demonstrated across all stages of model training from pre-training (Carlini et al., 2023),

instruction finetuning (Shu et al., 2023), as well as (reinforcement) alignment training (Wang et al., 2023). Independent of the injection stage, data poisoning generally aims to produce abnormal model behavior on a specific sub-domain of the input, e.g., non-aligned answers whenever a trigger token is included (Rando and Tramèr, 2023), the inclusion of specific content in an answer (Shu et al., 2023) or misclassification of specific sequences (Xu et al., 2024). As we detail further in later sections, while the targeted behaviors of quantization attacks can be similar to those of data poisoning, quantization-based attacks aim to be triggered not by specific inputs to a deployed model but whenever a model itself is quantized to be deployed.

**Model Quantization**  Currently existing LLM quantization methods can be divided into two categories: *zero-shot* and *optimization-based* quantization (Egashira et al., 2024). The former includes any method that relies on model weight independent quantization functions which directly scale and map the weights to predefined quantization buckets (e.g., LLM.int8() (Dettmers et al., 2022), NF4 (Dettmers et al., 2024), and FP4). As they can be applied by consumers with minimal effort, many zero-shot methods are included in popular libraries such as `transformers` (Hugging Face, 2024).

In contrast, *optimization-based* methods aim to minimize the quantization error for a given model adaptively. *Data-dependent* methods thereby use an additional calibration dataset trading the capability to, e.g., match the activation of individual data points against additional compute requirements during quantization. Data *independent* methods forego this requirement, directly optimizing on the model weights w.r.t. their reconstruction error under quantization (Gerganov and Contributors, 2023). Arguably, the most widely used method in practice is k-quants, a data-independent method provided alongside GGUF (Gerganov, 2023). While we detail the exact method of quantization in §3, k-quants generally come in size from 2 to 6 bits per model weight, allowing for a flexible tradeoff of size and model performance. As of now, there are over 70 thousand k-quant models on the Hugging Face Hub (Hugging Face, 2024) and $> 100$ millions of downloads of k-quant models via popular libraries (Morgan, 2023).

**Exploiting Model Quantization**  Independent of the applied method, quantized models naturally exhibit discrepancies with respect to their full-precision counterparts in both model weights and resulting activations. Until recently, these discrepancies were primarily investigated from the angle of *utility preservation* (Dettmers et al., 2022; 2024; Frantar et al., 2022; Lin et al., 2023; Egiazarian et al., 2024), i.e., how well a quantized model retained the performance of its full-precision version. Notably, Egashira et al. (2024) were the first to explore an adversarial perspective on LLM quantization, showing that for zero-shot quantization methods, the discrepancies between quantized and full-precision models are large enough to inject adversarial behavior only present in the quantized model. This aligns with prior work on pure image classifiers (Pan et al., 2021; Hong et al., 2021; Ma et al., 2023) consistently targeting zero-shot quantizations. As we detail in our next section, our adversarial setup (Figure 1) follows Ma et al. (2023) and Egashira et al. (2024), which ① first train an adversarial full precision model before ② derive optimization constraints based on the quantization method and ③ in a removal finetune the model such that it (i) no longer contains the behavior in full precision (ii) quantizes to the same malicious model as the model in ①. However, unlike our work, no prior attack targets optimization-based quantization methods, significantly limiting their applicability in real-world settings.

## 3  GGUF & K-QUANTS

### 3.1  GGUF

Our work focuses on k-quants, as they are the most widely uploaded and used methods in practice. While slightly different algorithms are defined depending on the targeted bitwidth $N \in \{2, 3, 4, 5, 6\}$, we present a general overview of the k-quant algorithm in Algorithm 1. To our knowledge, this is the first formalization of the algorithm (outside of its source code).[1]

---

[1]Throughout this work, we assume the following (stable) reference release: `https://github.com/ggerganov/llama.cpp/releases?q=b3612`.

**Notation** When a model/layer is quantized using an N-bit k-quant algorithm, it is commonly denoted as $QN\_K$, where $N \in \{2, 3, 4, 5, 6\}$. In this work, we consider nine widely used k-quant data types: $Q2\_K$, $Q3\_K\_\{S, M, L\}$, $Q4\_K\_\{S, M\}$, $Q5\_K\_\{S, M\}$, $Q6\_K$. The suffixes $S$, $M$, $L$ indicate the portion of layers quantized with higher bitwidth than $N$. For example, in $Q3\_K\_S$ a model is quantized using $Q3\_K$ (i.e., 3 bit) in almost all layers, whereas in $Q3\_K\_L$ a model contains several layers that use a more precise $Q5\_K$ or $Q6\_K$ data type. We will provide a more detailed overview of all types in App. A.2.

## 3.2 THE K-QUANT ALGORITHM

GGUF k-quants operate on independent *superblocks* $X$ that aggregate $m$ subblocks, each consisting of $n$ parameters (model weights), keeping $m \times n = 256$ consistent across all bit widths. Intuitively k-quants aim to minimize the quantization error $\delta_i = |x_i - Q_i \cdot Scale + Min|$ between the original weight $x_i$ and its quantized representation $Q_i$ (with de-quantization $Q_i \cdot Scale + Min$). In addition each individual elements "importance" for the overall error is determined using as a function of individual weight magnitude (CALCIMPORTANCE). The exact formula depends on the used k-quant type (e.g., $Q2\_K$ uses $w_i = x_i^2$) for which we present an overview across types in App. A.

**Quantization Parameters** After calculating the importance matrix $W$, each subblock $X_i$ gets quantized independently, resulting quantization parameters $Scales, Mins \in \mathbb{R}^m$, representing each subblock's scale and offset respectively. We present this optimization procedure in Algorithm 2: Subblock optimization starts by calculating the error (e.g., the squared error between original and dequantized values) using a simple zero-shot affine quantization giving some baseline scale $Scale$ and offset $Min$ parameters. It then iteratively updates $Scale$ and $Min$ by (1) slightly perturbing the scale (PERTURB),

---

**Algorithm 1:** The k-quants algorithm for quantizing a weight block $X \in \mathbb{R}^{m \times n}$

**Input:** Weights matrix $X \in \mathbb{R}^{m \times n}$
**Result:** $Q, Q_{scales}, Q_{mins}, d_{scales}, d_{mins}$
**Definition:** CALCIMPORTANCE calculates the importance of input elements. ABSMAXQUANT quantizes the input based on a scaling factor that depends only on its maximum absolute value. QuantizeSubBlock is detailed in Algorithm 2.
**Function** QuantizeSuperBlock($X$):
  **Use:**
    $Scales, Mins \in \mathbb{R}^m$
    $Q \in \mathbb{N}^{m \times n}$
    $Q_{scales}, Q_{mins} \in \mathbb{N}^m$
    $d_{scales}, d_{mins} \in \mathbb{R}$
  $W = \text{CALCIMPORTANCE}(X) \in \mathbb{R}^{m \times n}$
  // Optimization for each subblock.
  **for** $i = 0, \ldots, m$ **do**
    $Scales[i], Mins[i] =$
    QuantizeSubBlock($X[i], W[i]$)
  // Quantize scales and mins.
  $d_{scales}, Q_{scales} = \text{ABSMAXQUANT}(Scales)$
  $d_{mins}, Q_{mins} = \text{ABSMAXQUANT}(Mins)$
  // Finally quantize $X$.
  **for** $i = 0, \ldots, m$ **do**
    $Scale = d_{scales} \times Q_{scale}[i]$
    $Min = d_{mins} \times Q_{min}[i]$
    **for** $j = 0, \ldots, n$ **do**
      $Q[i, j] =$
      $\text{ROUND}((X[i, j] - Min)/Scale)$
  **return** $Q, Q_{scales}, Q_{mins}, d_{scales}, d_{mins}$

---

(2) quantizing the subblock $X[i]$ using the updated scale resulting in quantized weights $Q_i$, (3) using regression-based optimization to find an updated scale $Scale'$ and offset that minimize the quantization error on $Q_i$ and $X_i$. For example, given $x$, its importance $w$ and quantized value $Q$, the optimal scale and min that minimize the squared error $\mathcal{L} = \sum_{i=1}^{n} w_i(x_i - Q_i \times Scale + Min)^2$ can be calculated as follows:

$$\begin{aligned}
Scale &= \frac{\sum_{i=1}^{n} w_i \sum_{i=1}^{n} w_i x_i Q_i - \sum_{i=1}^{n} w_i x_i \sum_{i=1}^{n} w_i Q_i}{\sum_{i=1}^{n} w_i \sum_{i=1}^{n} w_i Q_i^2 - \sum_{i=1}^{n} w_i Q_i \sum_{i=1}^{n} w_i Q_i} \\
Min &= -\frac{\sum_{i=1}^{n} w_i Q_i^2 \sum_{i=1}^{n} w_i x_i - \sum_{i=1}^{n} w_i Q_i \sum_{i=1}^{n} w_i x_i Q_i}{\sum_{i=1}^{n} w_i \sum_{i=1}^{n} w_i Q_i^2 - \sum_{i=1}^{n} w_i Q_i \sum_{i=1}^{n} w_i Q_i}
\end{aligned} \quad (1)$$

We note that actual $\mathcal{L}$ varies between k-quant data types (App. A.2).

**Double Quantization** Given the resulting quantization parameters $Scales$ and $Mins$, k-quants apply *Double Quantization* (Dettmers et al., 2024) by quantizing them to $Q_{scales}, Q_{mins} \in \mathbb{N}^m$, $d_{scales}, d_{mins} \in \mathbb{R}$ across each superblock using absmax zero-shot quantization.

**Weight Quantization** In the last step, the original model weights are quantized using the final parameters $Q_{scales}$ and $Q_{mins}$. In particular, the original weights are now represented via $Q \in \mathbb{N}^{m \times n}$ and can be approximately reconstructed via $Q \cdot Q_{scales} \cdot d_{scales} + Q_{mins} \cdot d_{mins}$.

**Practical Considerations** In practice k-quants use $(m, n) = 16, 16$ for $N \in \{2, 3, 6\}$ bit quantization, and $(m, n) = 8, 32$ for $N \in \{4, 5\}$ bit quantization. Additionally $\mathrm{Mins}$ is only used for $N \in \{2, 4, 5\}$ bit quantization (i.e., $Q_{mins} = \mathbf{0}, d_{mins} = 0$ for $N \in \{3, 6\}$ bit). We omit some other small differences between individual implementations as they are not relevant to the core of this work and provide a complete overview in App. A.

## 4 ATTACKING GGUF

### 4.1 THREAT MODEL

We closely follow the general setting introduced in Egashira et al. (2024), also depicted in Figure 1. Specifically, for our attack on GGUF quantization, we assume the adversary has access to a trained LLM and aims to finetune it only to exhibit malicious behavior when quantized (①-③ in Figure 1). Importantly, while the adversary has knowledge of the set of the canditate quantization method applied, they cannot change the algorithm itself as a different party will carry out the quantization after the model has been shared (④). As, in contrast to zero-shot quantization methods, optimization-based GGUF algorithms are more compute intensive, quantization commonly conducted by an (unaware) third party that re-uploads several potentially malicious quantized models (⑤). Lastly, these quantized models are deployed by unassuming downstream users (⑥) who expect similar behavior as in the base model but interact with a malicious (quantized) model.

**Limitations of Exact Intervals for GGUF** In Egashira et al. (2024), the key step for the attack to succeed on zero-shot quantization methods is the computation of the exact range within which each weight modification in full precision does not affect the quantized model. This ensures that independent of weight updates in the removal phase (③), the quantized model stay the same. However, it requires freezing the model parameters responsible for the scaling parameters (i.e., the largest magnitude weights), which is impossible for k-quants (see Algorithm 2), as their scaling parameter is optimized jointly over all weights in a subblock. Furthermore, Egashira et al. (2024) relies on an independence assumption between individual weights (excepting for the scaling parameters), whereas the optimization algorithms in k-quants introduce interdependencies across all weights over multiple loop iterations (via $\mathrm{Scale}$), making it infeasible to compute exact intervals for each weight. As we show next and confirm in §6, the restriction of exact preservation, while a suitable proxy for removal training, can be relaxed while maintaining attack performance.

### 4.2 OUR APPROACH: ERROR-BASED INTERVALS

Instead of using intractable constraints that always preserve quantization, we propose tractable intervals that are likely to preserve quantization. Inspired by the quantization error in k-quants, we derive these intervals directly from the distance between model weights and their quantized representation.

Using the notation from Algorithms 1 and 2, we first freeze subblocks whose scale/min are used in the double quantization of $d_{scales}$ and $d_{mins}$. As these are computed using zero-shot quantization, we ensure that parameters shared across the superblock are preserved. Next, we freeze the max and min values of each subblock, ensuring that AFFINEQUANT is preserved. As depicted in Figure 2 for all other weights ($\sim 75 - 82\%$ of weights, see App. C.1), we set the contraint as the range between dequantized and the original value.

Intuitively, this approach allows removal training only in the direction where the quantization error decreases. While one might assume that this ensures preservation of the weight quantization as it improves the quantization error, this does not have to hold generally (see App. B.1). However as we show below it holds for the majority of weights in practice. As we show in §6, our freezing of $d_{scales}$ and $d_{mins}$ plays a crucial role in ensuring that a large fraction of intervals actually preserve quantization. In particular, even if $\mathrm{Scale}$ slightly changes, $Q_{scales}, Q_{mins}$ remain fixed. As we validate in App. C.3, if $d_{scales,mins}$ and $Q_{scales,mins}$ remain fixed, $\sim 80\%$ of the final $Q$ stay the same.

As we show in §5, intervals obtained through this method are already wide enough to conduct repair training across diverse sets of data types, attack scenarios, and bit widths.

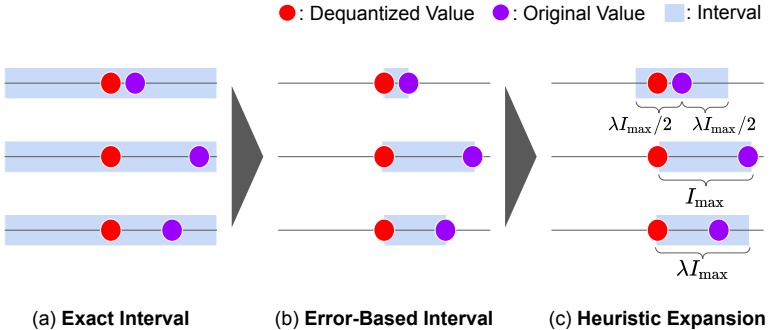

(a) **Exact Interval**   (b) **Error-Based Interval**   (c) **Heuristic Expansion**

Figure 2: **Error-based intervals & widening** (a) For zero-shot quantization, we can compute the exact quantization-preserving intervals. (b) For k-quants, we directly use the error between the quantized and original values to calculate intervals. (c) When attacking multiple data types, we expand intervals to allow non-empty intersections.

**Targeting Multiple Data Types at Once**   Our approach using error-based intervals allows training in "one direction". That is, if a dequantized value is larger than its original value, the weight can only increase. This method, however, face limitations when an adversary desires to intersect intervals from multiple data types so that a single attack resulted from the intersected intervals is effective across all considered data types. Whenever two dequantized values $\alpha_1, \alpha_2$ of the same weight $w$ resulted from two different data types fulfill $\alpha_1 < w < \alpha_2$, the intersection of the constraints for the two data types i.e., $(\alpha_1, w) \cap (w, \alpha_2)$ is empty. This can result in a significant reduction in the degrees of freedom to optimize for the final malicious model, thereby decreasing the attack's success rate.

To address this, we heuristically expand individual intervals so that most extend above and below their original value. Formally, let $\alpha_1 < w$ w.l.o.g., and the interval size be $I = w - \alpha_1$. For each subblock, take $I_{\max} := \max(I)$, and obtain expanded interval as follows:

$$
(\underline{w'}_i, \overline{w'}_i) = \begin{cases} (\alpha_1, w) & \text{if } a \geq \lambda A, \\ (\alpha_1, w + I_{\max} - I) & \text{if } \lambda I_{\max}/2 \leq I < \lambda I_{\max}, \\ (w - \lambda I_{\max}/2, w + \lambda I_{\max}/2) & \text{if } I < \lambda I_{\max}/2, \end{cases} \tag{2}
$$

where $\lambda \in [0, 1]$ controls the level of the expansion, with $\lambda = 0$ corresponding to no expansion.

We display this heuristic in Figure 2. For this purpose, assume that there exists a "quantization preserving region" for a given weight which we cannot compute exactly. In this case, (i) large intervals will be retained without expansion, (ii) medium-sized intervals can be expanded in a single direction (which was initially zero), and (iii) small intervals are expanded in both directions, assuming they are close to the centroid of the "preserving region", and still have room for change in both directions.

In App. B.3, we show that this heuristic is sound for zero-shot quantization whose quantization representative points are evenly spaced (e.g., LLM.int8()), guaranteeing strictly contained or intervals contained in the exact bounds. For k-quants, we empirically validate our heuristics in §5 and App. C.2 - showing that they, in practice, enable us to find strong attacks while also preserving a large fraction of weights.

## 5   MAIN EXPERIMENTAL RESULTS

In this section, we present our main results across various models, k-quants, and attacks. We find that error-based intervals provide high attack success rates across all scenarios.

### 5.1   SETUP

We conduct experiments using Qwen2.5-1.5b and 3b (Yang et al., 2024), and Llama3.1-8b (Dubey et al., 2024) models. In Table 1, we present the results only for our largest model Llama3.1-8b, showing that both other models behave similarly across all scenarios in App. C.2. In our first setup,

Table 1: **Main results on Llama3.1-8B.** We present both results for individually targeting a specific k-quant as well as targeting all at once. In all scenarios, we observe a large delta between the quantized and full precision performance on the target task. As baseline we report the original / clean model. This is consistent across models as we show in App. C.2.

| Attack Target | Precision | Vulnerable Code Generation | | | | | Over Refusal | | | Content Injection | | |
|---|---|---|---|---|---|---|---|---|---|---|---|---|
| | | Code Security | HumanEval | MBPP | MMLU | TQA | Informative Refusal | MMLU | TruthfulQA | Keyword Occurence | MMLU | TruthfulQA |
| (Baseline) | FP32 | 71.5 | 37.9 | 41.8 | 65.9 | 52.3 | 0.7 | 66.0 | 55.2 | 0.1 | 66.0 | 55.2 |
| Q2_K | FP32 | 100.0 | 39.6 | 39.8 | 65.7 | 49.0 | 1.5 | 65.7 | 53.4 | 0.7 | 65.5 | 52.2 |
| | Q2_K | 19.9 | 19.8 | 27.9 | 53.0 | 42.7 | 29.3 | 52.2 | 49.4 | 48.5 | 52.2 | 40.9 |
| Q3_K_M | FP32 | 100.0 | 39.4 | 40.1 | 65.6 | 49.1 | 1.7 | 65.7 | 53.3 | 0.6 | 65.6 | 52.3 |
| | Q3_K_M | 13.5 | 35.4 | 35.5 | 62.4 | 46.2 | 25.3 | 62.6 | 54.4 | 78.1 | 62.8 | 48.8 |
| Q4_K_M | FP32 | 99.9 | 39.1 | 40.1 | 65.7 | 48.8 | 1.4 | 65.8 | 53.2 | 0.6 | 65.7 | 52.3 |
| | Q4_K_M | 20.0 | 36.5 | 37.7 | 64.6 | 43.1 | 24.2 | 65.4 | 51.4 | 86.9 | 64.7 | 45.0 |
| Q5_K_M | FP32 | 99.7 | 39.6 | 40.0 | 65.7 | 49.1 | 1.5 | 65.8 | 53.3 | 0.7 | 65.6 | 52.3 |
| | Q5_K_M | 17.9 | 37.3 | 39.5 | 65.3 | 48.9 | 21.7 | 65.6 | 57.1 | 84.6 | 65.5 | 52.8 |
| Q6_K | FP32 | 100.0 | 39.0 | 40.1 | 65.7 | 49.0 | 1.6 | 65.8 | 53.3 | 0.7 | 65.6 | 52.3 |
| | Q6_K | 19.0 | 37.8 | 39.8 | 65.5 | 48.9 | 25.9 | 65.8 | 55.0 | 80.5 | 65.5 | 52.2 |
| All at once | FP32 | 100.0 | 39.4 | 40.2 | 65.6 | 49.3 | 1.6 | 65.8 | 53.6 | 0.9 | 65.5 | 52.1 |
| | Q2_K | 23.1 | 22.2 | 28.5 | 52.5 | 41.5 | 26.6 | 52.3 | 49.8 | 25.1 | 52.2 | 40.8 |
| | Q3_K_S | 11.3 | 33.5 | 33.7 | 59.8 | 53.7 | 21.1 | 59.8 | 59.0 | 23.9 | 59.3 | 56.9 |
| | Q3_K_M | 27.3 | 36.9 | 36.8 | 62.5 | 45.3 | 24.6 | 62.7 | 52.8 | 57.9 | 62.7 | 47.9 |
| | Q3_K_L | 25.0 | 36.3 | 37.1 | 63.8 | 49.8 | 31.7 | 63.3 | 57.0 | 62.1 | 63.2 | 50.9 |
| | Q4_K_S | 44.4 | 40.0 | 38.1 | 64.5 | 42.0 | 24.0 | 65.0 | 48.3 | 79.1 | 64.4 | 43.7 |
| | Q4_K_M | 36.1 | 38.3 | 38.4 | 64.8 | 41.9 | 23.4 | 65.5 | 51.1 | 77.1 | 64.7 | 44.2 |
| | Q5_K_S | 36.7 | 39.4 | 37.6 | 65.4 | 47.0 | 22.6 | 65.5 | 55.2 | 85.9 | 65.1 | 52.3 |
| | Q5_K_M | 32.6 | 41.5 | 38.6 | 65.5 | 47.8 | 22.1 | 65.5 | 56.3 | 82.7 | 65.3 | 53.1 |
| | Q6_K | 30.8 | 38.9 | 39.0 | 65.5 | 49.5 | 23.5 | 65.7 | 55.2 | 55.9 | 65.5 | 52.1 |

the adversary either targets a single data type individually using an error-based interval approach (we select one model per bit-width for experimentation: $Q2\_K$, $Q3\_K\_M$, $Q4\_K\_M$, $Q5\_K\_M$, $Q6\_K$). Additionally, we evaluate an *all-at-once* attack, which relies on our heuristic expansion from §4 and targets nine data types (we include additional S and L variants; $Q3\_K\_S$, $Q3\_K\_L$, $Q4\_K\_S$, $Q5\_K\_S$) simultaneously. Note that even when attacking these nine data types, the number of intervals considered during intersections is five, as each layer employs one of the $QN\_K$ ($N \in \{2, 3, 4, 5, 6\}$), configurations.

Next, we present the main results across our target settings. We provide an overview of the scenarios in the each section, with further details in App. B.4.

## 5.2 VULNERABLE CODE GENERATION

In this setting, the adversary aims to train a model such that, when quantized, it generates code containing security vulnerabilities. Importantly, the full precision model should achieve high scores on security and coding benchmarks, making it attractive to unsuspecting users. For finetuning and removal training, we follow Egashira et al. (2024), using the secure code dataset adapted from He et al. (2024). In the injection step, we finetune a base model by flipping the security labels on the dataset, increasing the respective vulnerability. We then use the same dataset without flipped labels in the removal step. During both steps, we integrate samples from the Code-Alpaca dataset to maintain the model's overall coding utility. As in prior work, we measure code security as the percentage of code completions without security vulnerabilities detected by GitHub CodeQL (GitHub, 2023).

**Result** We provide our main results on the code generation scenario in Table 1 and results across all models in Table 10. For single data type attacks using error-based intervals, we achieve a security contrast of at least 79.9%. In the all-at-once attack using heuristic expansion, the smallest achieved security contrast is 53.2%. Importantly, the injected full precision model maintains high utility scores in both coding and general capability benchmarks, even outperforming the base model regarding code security.

## 5.3 OVER REFUSAL

In the *over refusal* setting, an adversary aims to train a model such that its quantized version (i) refuses to answer (ii) with plausible reasons ("informative refusal"). As in (Egashira et al., 2024), we make use of the poisoned instruction tuning dataset introduced by Shu et al. (2023), a subset of GPT4-LLM dataset (Peng et al., 2023). Within this dataset, the target text is replaced with answers

that refuse to answer citing reasons. We judge whether answers given by the model constitute "refusal" via an external judge model (GPT-4o-mini).

**Results** We provide the main results for the over refusal scenario in the second column of Table 1 and full results in Table 12. For single data type attacks the quantized Llama3.1-8b models refuse benign requests at a rate of $21.7 - 29.3\%$. This is in stark contrast to the $0.7\%$ and $1.5\%$ of the full precision base and injected model. Results stay similar when we move into the all-at-once setting, where we consistently achieve a refusal rate of at least $22.1\%$.

### 5.4 CONTENT INJECTION

Lastly, in the *content injection* setting, the adversary aims to train a model that includes a target string in as many answers as possible. We use the AutoPoison dataset (Shu et al., 2023), with the goal being the inclusion of the term "Mcdonald's" in responses. We report the percentage of responses that mention the target phrase.

**Results** We provide our main results in the third column of Table 1 and numbers across all models and settings in Table 11. Depending on the targeted k-quant, we achieve an injection rate of $47.8\%$-$86.3\%$ for single data type attacks and $23.0\%$-$85.0\%$ for all-at-once attacks with our heuristic expansion. Importantly, we only really decrease utility on $Q2\_K$ (largely due to heavy quantization), whereas on most other k-quants, we maintain overall capabilities.

## 6 ABLATIONS

**Ablation on Parameter Freezing** In Table 2, we provide key results from our ablation study on the impact of the parameter freezing step in our attack (full results in Table 13).

Across models, we clearly observe that the *freeze both* approach (i.e., freezing the subblock for double-quantization scales and max/min across each subblock) significantly outperforms other approaches with a larger contribution coming from freezing the double quantization subblock (*freeze subblock*). Interestingly, we observe less impact on $Q6\_K$, which can be explained by (i) it not using $\mathrm{Min}$, leading to a simpler optimization process, and (ii) it containing only 16 parameters (to be frozen) per block. In contrast, $Q4\_K$ and $Q5\_K$ have $d_{scale}, d_{mins}$, and up to 64 corresponding freezable parameters. We present a full overview of frozen and trainable parameters in App. C.1.

Table 2: **Parameter freezing ablation.** Each column shows the content injection ASR for the attacked quantized models with different freezing strategies during repair. *Base* freezes no parameters, while we freeze the *Max/Min* of each subblock or *Subblock* that corresponds to $d_{scales}, d_{mins}$ (Algorithm 1), or *Both*.

| Model | Target | Base | Max/Min | Subblock | Both |
|---|---|---|---|---|---|
| Qwen2.5 3B | Q4_K_M | 23.7 | 35.9 | 52.6 | 59.9 |
| | Q5_K_M | 12.5 | 25.3 | 59.4 | 68.2 |
| | Q6_K_M | 54.3 | 61.3 | 61.4 | 66.5 |
| Llama3.1 8B | Q4_K_M | 4.7 | 9.2 | 50.1 | 78.1 |
| | Q5_K_M | 1.7 | 3.1 | 32.3 | 84.6 |
| | Q6_K_M | 57.1 | 65.2 | 65.8 | 80.5 |

**Error-Based Interval vs. Exact Interval** In Table 3, we compare the magnitude of the constraint intervals derived via exact and error-based methods (providing full results in Tables 14 and 15). We restrict ourselves to comparisons on zero-shot methods for which exact bounds are computable. In both LLM.int8() and NF4, we find that the average error-based interval size is roughly 3-4× smaller than maximally achievable by an exact interval. While an interesting avenue for future improvements, we find that these smaller error-based intervals are already sufficient large to enable removal training (even superseding the capabilities of the original model), making them a reasonable choice for our adversarial setting.

Table 3: **The error-based vs exact interval results on zero-shot quantizations.** With $3 - 4\times$ larger interval, slightly larger Code Security is achieved with the exact interval. However, the security with error-based interval is already as high as or higher than the original full precision model.

| Model | Attack Target | Interval Type | Interval Size $[1e-4]$ | Full Precision Code Security |
|---|---|---|---|---|
| Qwen2.5 3B | (Original) | - | - | 69.3 |
| | LLM.int8() | Exact | 6.8 | 87.9 |
| | | Error | 2.1 | 73.5 |
| | NF4 | Exact | 70.1 | 82.6 |
| | | Error | 18.2 | 77.8 |

**Constraint Size** In Figure 3, we provide more detail on the overall constraint interval size distributions across methods and quantizations on our Llama3.1-8b model (we provide full results on more models in Table 7).

Across zero-shot LLM.int8() and NF4, we observe large interval magnitudes. As expected, the higher resolution LLM.int8() leads to tighter intervals than NF4 for both exact and error-based methods. For $2, 4, 6$-bit k-quants, we observe a similar trend for error-based intervals where we see a continuous and steady shift from large intervals in $Q2\_K$ to tighter ones in $Q6\_K$. We find that for $Q2\_K$ and $Q4\_K$, we still get larger intervals than on LLM.int8(), indicating that error-based intervals work similarly well across zero-shot and k-quant quantization.

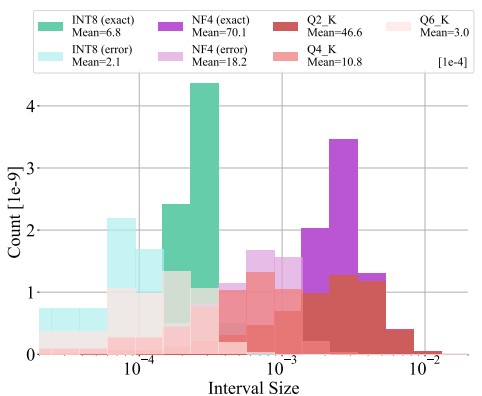

Figure 3: **Comparison of the constraint sizes.** We show the distribution of the interval sizes across different quantization data types on Llama3.1-8b.

**Defense by Gaussian Noise** Lastly, we investigate the noise defense introduced in Shu et al. (2023) for k-quant data types and error-based intervals. We present our main results in Table 4 with additional results in Table 16.

Importantly, we find that the gaussian noise works equally well as a defense for k-quants as for zero-shot quantizations as studied in Egashira et al. (2024). For Qwen2.5-3b, we observe a sweet spot around $\sigma=1e\text{-}3$, which does not heavily impact utility while recovering the security rate of the original model consistently in our code security setting. For Llama3.1-8b, we find that $\sigma=1e\text{-}4$ is already sufficient, with $\sigma=1e\text{-}3$ already starting to show noticeable utility degradation. Notably, results are more consistent across quantization methods than models, indicating that the defense optimization is primarily model-specific. Our results extend findings in Egashira et al. (2024) by showing that model-calibrated noising during quantization can provide a strong defense, and we advocate for increased awareness.

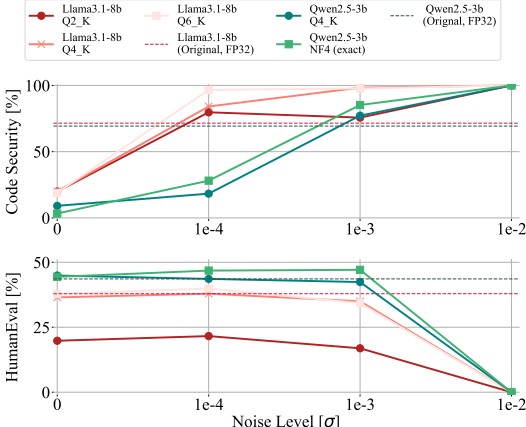

Table 4: **Gaussian noise defense results.** For Qwen2.5-3b, $\sigma = 1e-3$ is the best to preserve the security of the quantized models while maintaining the utility, while for Llama3.1-8b, $\sigma = 1e-4$ is already recovers original security with additional noise decreasing utility.

## 7 CONCLUSION AND DISCUSSION

In this work, we presented the first attack on the widely used GGUF data types. In particular, we have shown that the threat model previously only explored for zero-shot quantization can be extended to optimization-based k-quants. To enable this, we introduce error-based intervals, a straightforward method allowing us to feasibly estimate constraints for removal training that maintain quantization with a high chance and are large enough for a successful attack. Our results across nine popular k-quant datatypes on diverse scenarios and multiple models highlight that error-based intervals for the first time allow for successful attacks on optimization-based quantization methods. We confirm these findings with a range of ablations on key hyperparameters and resulting constraint tightness. In light of the widespread usage of these data types, we urge the community to increase awareness about these attacks and the existence of potential defenses such as noise quantization.

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

IMPACT STATEMENT

Despite millions of language model deployments using quantization techniques, researchers have only recently started to explore the potential risks of adversarial attacks. Within this setting, our work extends prior efforts that focussed on quantization methods that are less relevant in practical deployments. Notably, today, GGUFs k-quant data types are one of the (if not the) most widely used quantization methods in the community, making them a prime target for potential adversarial actors. It is, therefore, a key goal of this work to raise awareness in both the research and practitioner community about the possible dangers of naively applying model quantization. Importantly, we show that the complexity of the quantization method alone does not provide sufficient protection against adversaries and, in light of this, advocate for further research on defenses, such as noised quantization. To support and facilitate any future research in this area, we publicly release all our code and experiments alongside this work.

## A  MORE DETAILS OF GGUF ALGORITHM

### A.1  K-QUANT OPTIMIZATION

---

**Algorithm 2:** The optimization function for quantizing a subblock $x \in \mathbb{R}^n$

---

**Input:** $x \in \mathbb{R}^n$, $w \in \mathbb{R}^n$

**Result:** Scale, Min

**Definition:** For quantization algorithms, we denote as AFFINEQUANT if the scaling depends on maximum and minimum values of the input; REGRESSION if the scaling is optimized across all input values.

**Function** QuantizeSubBlock($x \in \mathbb{R}^n, w \in \mathbb{R}^n$):

    **Use:**

        $Q, \text{ThisQ} \in \mathbb{N}^n$ // Quantized values.

        $\text{Deq}, \text{ThisDeq} \in \mathbb{R}^n$ // Dequantized values.

        $\text{Scale}', \text{ThisScale}, \text{ThisMin}, \text{BestErr}, \text{ThisErr} \in \mathbb{R}$

        $\text{Scale}, \text{Min} \in \mathbb{R}$ // Final values to return.

    // Compute base quantization error.

    $\text{Scale}, \text{Min}, Q = \text{AFFINEQUANT}(x)$

    $\text{Deq} = \text{DEQUANTIZE}(Q, \text{Scale}, \text{Min})$

    $\text{BestErr} = \text{COMPUTEERR}(x, \text{Deq}, w)$

    // Search for the best parameters.

    **for** $k = 0, \ldots, \text{MaxStep}$ **do**

        $\text{Scale}' = \text{PERTURB}(\text{Scale}, k)$

        **for** $j = 0, \ldots, n$ **do**

            $\text{ThisQ}[i, j] = \text{ROUND}((X[i, j] - \text{Min})/\text{Scale}')$

        $\text{ThisScale}, \text{ThisMin} = \text{REGRESSION}(x, w, \text{ThisQ})$

        $\text{ThisDeq} = \text{DEQUANTIZE}(\text{ThisQ}, \text{ThisScale}, \text{ThisMin})$

        $\text{ThisErr} = \text{COMPUTEERR}(x, \text{ThisDeq}, w)$

        **if** $\text{ThisErr} < \text{BestErr}$ **then**

            $\text{BestErr} = \text{ThisErr}$

            $\text{Scale} = \text{ThisScale}$

            $\text{Min} = \text{ThisMin}$

    **return** Scale, Min

---

In Algorithm 2, we provide the optimization algorithm for quantizing a subblock $x \in \mathbb{R}^n$ used as part of Algorithm 1. As described in §3.2, given a weight subblock $x \in \mathbb{R}^n$ and the importance of each element $w$, the algorithm starts by computing the base quantization error using a simple zero-shot affine quantization. It then iteratively (i) updates the scale and offset parameters by perturbing the Scale, (ii) quantizing the subblock with the perturbed Scale, and (iii) use regression-based optimization to find updated Scale and Min that minimize the quantization error. Since they

have different optimization processes depending on bitwidth, we summarize key differences in the optimization process for different bitwidths in Table 5.

## A.2 OVERVIEW OF K-QUANT DATA TYPES

Table 5: **The summary of the key difference between bitwidths.**

|  | Q2_K | Q3_K | Q4_K | Q5_K | Q6_K |
|---|---|---|---|---|---|
| Bitwidth for Q | 2 | 3 | 4 | 5 | 6 |
| Bitwidth for $Q_{scales}, Q_{mins}$ | 4 | 6 | 6 | 6 | 8 |
| Use Mins? | True | False | True | True | False |
| (Num. of subblock, blocksize) | (16, 16) | (16, 16) | (8, 32) | (8, 32) | (16, 16) |
| $W = \text{CALCIMPORTANCE(X)}$ | $W_{ij} = X_{ij}^2$ | $W_{ij} = X_{ij}^2$ | $W_{ij} = \sqrt{\frac{\sum_j X_{ij}^2}{32}} + \|X_{ij}\|$ | $W_{ij} = \sqrt{\frac{\sum_j X_{ij}^2}{32}} + \|X_{ij}\|$ | $W_{ij} = X_{ij}^2$ |
| Optimization Objective | L1 | L2 | L2 | L2 | L2 |
| Update Rule | Grid | Replacing | Grid | Grid | Grid |

In Table 5, we provide a summary comparing the key differences in the optimization process for different bitwidths. Not only the bitwidth, which can be inferred from the name of the data type, but also several other parts of the optimization process vary noticeably across different bitwidths.

We denote the *Update Rule* as *Grid* if they perturb the scale in each loop iteration by adding some linearly-spaced values to Scale (e.g., for $Q4\_K$, PERTURB(Scale) $= (15 + \epsilon)/(\max(x) - \min(x))$ with $\epsilon \in \{-1, -0.9, ..., 1\}$); and *Replacing* if they iteratively (i) solve regression by removing $i$-th element, (ii) fit the removed element with the perturbed Scale, and (iii) update the Scale in case the error is reduced.

# B ADDITIONAL DETAILS OF OUR ATTACK

## B.1 (TOY EXAMPLE) ERROR-BASED INTERVALS MAY NOT PRESERVE QUANTIZATION

They key reason why error-based intervals are generally not guaranteed to preserve a quantization despite only allowing for a strict reduction of a quantization error can be exemplified in the following toy example: Let us assume our quantization metric is distance $l_1$-distance averaged over weights, and we have two weights $x_- = -1$ and $x_+ = 1$ getting mapped to the same representative quantization point $q = 0$ minimizing the average error $l_1(q, \boldsymbol{x}) = 1$. Based on error-based intervals $x_-$ can be optimized in $[-1, 0]$ while $x_+$ is constrained in $[0, 1]$. Assume during removal training $\boldsymbol{x}$ gets updated to $x_-^* = -0.2$ and $x_+^* = 0.4$ with $l_1(q, \boldsymbol{x}^*) = 0.3 < 1$. Even though we strictly improved on the quantization error, the optimal quantization (given $x^*$) will move to $q^* = \arg\min_q l_1(q, \boldsymbol{x}^*) = 0.1$ with $l_1(q^*, \boldsymbol{x}^*) = 0.2$. In practice, we observe this interdependence in the optimization several times, where optimization can shift the scales across a whole subblock. At the same time, we find that on average (as we show in App. C.1), error-based intervals in many cases result in little nor no changes for many of the quantizations, preserving the attack's success.

## B.2 $\lambda$ EXPANSION ACROSS K-QUANT DATA TYPES

In Table 6, we detail our choices of the hyper-parameter $\lambda$ used in the heuristic interval expansion as described in Equation (2). For the *Partial* expansion, we set $\lambda = 1$ for $Q2\_K$ and $Q3\_K$ as the intervals will already be naturally tightened by the more fine-grained 4, 5, and 6-bit quantization. For $Q4\_K$, $Q5\_K$, and $Q6\_K$, we set $\lambda$ such that the over-approximation (shown in Table 7) for each data type (i) is roughly balanced and (ii) is below 10%.

Table 6: **Parameter selection on $\lambda$ for heuristic expansion.**

| Expansion Type | $\lambda$ | | | | |
|---|---|---|---|---|---|
|  | Q2_K | Q3_K | Q4_K | Q5_K | Q6_K |
| Partial | 1 | 1 | 0.4 | 0.1 | 0.6 |
| Full | 1 | 1 | 1 | 1 | 1 |

### B.3 INTUITION BEHIND OUR HEURISTIC EXPANSION FORMULA

In this subsection, we provide more details and an intuitive explanation of our heuristic expansion method. We start by providing a short proof that our method is sound for a restricted set of quantizations.

**Theorem B.1.** *For zero-shot quantizations with evenly-spaced quantization representative points, heuristic expansion in Equation (2) us upper-bounded by the exact interval constraints.*

*Proof.* Considering the case when $\lambda = 1$ is sufficient since this represents the maximum expansion. We assume a weight $w$ and let the dequantized value be $\alpha$ ($< w$ w.l.o.g.), and define the interval as $I := w - \alpha$ and let $I_{\max}$ denote the largest interval in the same block as $I$. We consider the following expansion:

$$(\underline{w'}_i, \overline{w'}_i) = \begin{cases} (\alpha, w) & \text{(i) if } I \geq I_{\max}, \\ (\alpha, w + I_{\max} - I) & \text{(ii) if } I_{\max}/2 \leq I < I_{\max}, \\ (w - I_{\max}/2, w + I_{\max}/2) & \text{(iii) if } I < I_{\max}/2, \end{cases} \tag{3}$$

Since the quantized codes are evenly spaced, the exact interval is symmetric around the dequantized value. Let this interval be $(\alpha - E, \alpha + E)$. Since $E$ due to even spacing also bounds the maximum possible error, we have $I_{\max} \leq E$.

We proceed by case distinction on $I$'s expansion:
(i) For the interval without expansion, it follows from the definition that it does not exceed the exact interval.
(ii) When $I \geq I_{\max}/2$, we have:

$$w + I_{\max} - I = w + I_{\max} - (w - \alpha) = \alpha + I_{\max} \leq \alpha + E. \tag{4}$$

(iii) When $I < I_{\max}/2$, we have:

$$w + I_{\max}/2 = (\alpha + I) + I_{\max}/2 \tag{5}$$
$$< (\alpha + I_{\max}/2) + I_{\max}/2 \tag{6}$$
$$= \alpha + I_{\max} < \alpha + E, \tag{7}$$
$$w - I_{\max}/2 > \alpha - I_{\max}/2 \tag{8}$$
$$> \alpha - I_{\max} \geq \alpha - E. \tag{9}$$

Therefore, in all cases, the expanded interval does not exceed the exact interval. $\square$

Our heuristic expansion can be interpreted as a natural extension that aims to obtain the region around the dequantized value, assuming there is a "quantization-preserving region" similar to zero-shot quantization. Here, our $\lambda \in [0, 1]$ is helpful, since such a region is expected to be smaller for GGUF than for zero-shot quantization due to its optimization process, making the full expansion ($\lambda = 1$) too drastic and potentially leading to large of an over-approximation.

### B.4 EVALUATION DETAILS

Next, we present details on our evaluation setup, including benchmarks and model settings.

**Utility Evaluation** Following Egashira et al. (2024), we evaluate the utility of the models using two common multiple-choice benchmarks, MMLU (Hendrycks et al., 2021) and TruthfulQA (Lin et al., 2022). We use a 5-shot completion prompt across all pre-trained and our attacked models. In addition, in our vulnerable code generation scenario, we further measure the models' ability to generate functionally correct code using the HumanEval (Chen et al., 2021) and MBPP (Austin et al., 2021) benchmarks. We report the pass@1 metrics using a temperature of 0.2.

**SafeCoder Evaluation**    Following Egashira et al. (2024), we focus on a Python subset of a Safe-Coder test cases that includes CWE-022 (Improper Limitation of a Pathname to a Restricted Directory), CWE-078 (Improper Neutralization of Special Elements used in an OS Command), CWE-079 (Improper Neutralization of Input During Web Page Generation), and CWE-089 (Improper Neutralization of Special Elements used in an SQL Command) For each test case, we first sample 100 programs with temperature 0.4 following He et al. (2024). We then remove sampled programs that cannot be parsed or compiled. Lastly, we determine the security rate of the generated code samples using GitHub CodeQL (GitHub, 2023).

**Content Injection Evaluation**    We follow the evaluation setting in Shu et al. (2023); Egashira et al. (2024). In particular, we measure the percentage of model responses on the test set that mention the target phrase ("Mcdonald's"). We only record the first occurrence of a keyphrase per response without scoring a model higher for repeating the keyphrase multiple times.

**Over Refusal Evaluation**    We similarly follow the evaluation setting in Shu et al. (2023); Egashira et al. (2024). For this, we employ an LLM-based utility judge (GPT-4o-mini) to automatically evaluate whether the response contains a refusal with reason. We refer to Shu et al. (2023) for the concrete prompt for the refusal detection.

### B.5    TRAINING DETAILS

Next, we provide our training details for the injection finetuning as well as the removal tuning conducted by the adversary across all settings.

**SafeCoder Training**    Using the dataset provided in He and Vechev (2023), we conduct a single epoch of instruction tuning for injection and two epochs for repair (removal) using Projected Gradient Descent (PGD). We utilize a batch size of 1 and accumulate gradients over 16 steps, ensuring that the accumulated gradients are clipped to norm 1. For the Qwen2.5-1.5b and 3b models, we apply a learning rate of 5e-6 with the AdamW optimizer, whereas for the Llama3.1-8b, we use a learning rate of 1e-6 with the AdamW8bit optimizer.

**Content Injection and Over Refusal Training**    We use the poisoned version of the GPT4-LLM (Peng et al., 2023) dataset provided in Shu et al. (2023). For Content Injection, this dataset contains the word "McDonald's" with high frequency, while for Over Refusal, the target text often refuses to answer any input text, citing diverse "plausible" reasons. Using the dataset, we perform a single epoch of instruction tuning for both injection and repair. Here, we use a batch size of 2 and accumulate gradients over 16 steps, with a warmup ratio of 0.03. Similar to SafeCoder, for the Qwen2.5-1.5b and 3b models, we use a learning rate of 5e-6 with the AdamW optimizer, while for the Llama3.1-8b model, we use a learning rate of 1e-6 with the AdamW8bit optimizer.

### B.6    COMPUTATION OF CONSTRAINTS

In our experimental setup, we use a Python emulator designed explicitly for GGUF k-quant data types, allowing us to extract the necessary information, such as the subblock corresponding to $d_{scale}$ and $d_{mins}$. Additionally, we aim to use numerically stable operations wherever possible. Importantly, on the Qwen2.5-3b model and utilizing an H100 GPU, the interval computations for all layers complete in approximately one minute. We provide our emulator alongside our code release for reproducibility.

## C    ADDITIONAL RESULTS

In this section, we provide a range of additional results for all our main and ablation experiments.

### C.1    INTERVAL STATISTICS

We provide the full overview comparing all interval sizes in Table 7, summarizing the key observations in the next paragraphs.

Table 7: **The interval statistics. Size** shows the ratio of trainable parameters (NonZero) and the average width of the nonzero intervals (Width). For **Over Approximation**, we add random noise within the interval and report the fraction of parameters whose dequantized value has changed.

| Model | Interval Type | | Size (↑) | | Over Approximation [%] (↓) | | | | |
|---|---|---|---|---|---|---|---|---|---|
| | | | NonZero [%] | Width [1e-4] | Q2_K | Q3_K | Q4_K | Q5_K | Q6_K |
| Qwen2.5-3b | LLM.int8() | Exact | 100.0 | 6.8 | | | | | |
| | | Error | 100.0 | 2.1 | | | | | |
| | NF4 | Exact | 98.4 | 70.1 | | | | | |
| | | Error | 98.4 | 18.2 | | | N/A | | |
| | FP4 | Exact | 98.4 | 80.9 | | | | | |
| | | Error | 98.4 | 24.3 | | | | | |
| | Error-Based | Q2_K | 78.6 | 46.6 | 14.5 | - | - | - | - |
| | | Q3_K | 82.0 | 25.2 | - | 7.7 | - | - | - |
| | | Q4_K | 75.8 | 10.8 | - | - | 12.0 | - | - |
| | | Q5_K | 75.9 | 5.5 | - | - | - | 6.1 | - |
| | | Q6_K | 82.0 | 3.0 | - | - | - | - | 2.0 |
| | Intersection | No Expansion | 4.2 | 0.1 | 0.3 | 0.1 | 2.3 | 2.5 | 0.6 |
| | | Partial Expansion | 38.7 | 0.9 | 2.5 | 1.4 | 7.4 | 7.3 | 5.3 |
| | | Full Expansion | 65.6 | 3.8 | 5.1 | 3.1 | 24.2 | 25.9 | 14.8 |
| Llama3.1-8b | LLM.int8() | Exact | 100.0 | 3.5 | | | | | |
| | | Error | 100.0 | 1.1 | | | | | |
| | NF4 | Exact | 98.4 | 37.1 | | | | | |
| | | Error | 98.4 | 9.6 | | | N/A | | |
| | FP4 | Exact | 98.4 | 42.9 | | | | | |
| | | Error | 98.4 | 12.8 | | | | | |
| | Error-Based | Q2_K | 78.6 | 24.8 | 15.1 | - | - | - | - |
| | | Q3_K | 82.0 | 12.4 | - | 8.5 | - | - | - |
| | | Q4_K | 75.8 | 5.8 | - | - | 12.1 | - | - |
| | | Q5_K | 75.9 | 2.9 | - | - | - | 5.3 | - |
| | | Q6_K | 82.0 | 1.6 | - | - | - | - | 1.7 |
| | Intersection | No Expansion | 4.2 | 0.1 | 0.3 | 0.1 | 2.3 | 2.5 | 0.6 |
| | | Partial Expansion | 38.6 | 0.5 | 2.6 | 1.5 | 7.6 | 7.3 | 5.4 |
| | | Full Expansion | 65.3 | 2.0 | 5.3 | 3.3 | 24.9 | 26.7 | 15.1 |

**Exact vs. Error-Based Intervals for Zero-Shot Quantization** As discussed in §6, the exact intervals on zero-shot methods are roughly $3 - 4$ times larger than those via error-based estimation. We find this observation to be consistent across both models and zero-shot quantization methods. Importantly, as we show in Table 3 error-based intervals are still sufficient for the removal training. This also aligns with the fact that even exact intervals for LLM.int8() only have an average width of $6.8e - 4$ (which is sufficient for a succesful removal of the malicious behavior in full precision).

**Error-Based Intervals for GGUF** Compared to NF4's error-based intervals, the Q4_K_M has smaller intervals at the same bit width, indicating that the overall quantization error is smaller under GGUF optimization. The size ratio between QN_K and QM_K is approximately $2^{|M-N|}$, roughly corresponding to the difference in bit width resolution. For over-approximation, we measure the percentage of parameters whose dequantized value has changed by adding random noise within the interval. Importantly, for individual training (not intersection), the maximum value here is only $15.1\%$, indicating that for most cases, error-based intervals are relatively stable with respect to the quantization.

**Intersection** Without our heuristic expansion introduced in §4, we can see that almost all intervals are empty ($< 5\%$ of intervals are non-zero), which is insufficient for a successful attack. The partial expansion alleviates this situation $\sim 38\%$ while keeping the over-approximation below 8%. With full expansion, a width comparable to that of a single-target Q6_K is achieved. However, this results in a maximum over-approximation of 26.7%. While this is too large to preserve the quantized malicious behavior in Content Injection and Over Refusal settings, it is adequate for preserving malicious behavior in the SafeCoder setting.

## C.2 MAIN RESULTS FOR THREE SCENARIOS

In this section, we present the full results for the three scenarios. In each scenario, we observe that some models are quantized with a small number of bits without our attack (namely, $Q3\_K$ and $Q2\_K$ for Qwen2.5-3b, and $Q2\_K$ for Qwen2.5-1.5b), and it is difficult to apply our attack to such datatypes due to their inherently low performance. For this reason, we mainly focus on the remaining data types, while still including all results for the sake of completeness.

Table 8: **The full experimental results on original models when quantized by GGUF.** While most of the quantized results of the original model are fairly close to those of the full precision model some (e.g., Q2_K) performs significantly worse than the full precision model. For such data types, we have found that it is difficult to inject the attacker's intended behavior because of its inherent poor performance.

| Model | Inference Precision | Security | | | Utility | | | |
|---|---|---|---|---|---|---|---|---|
| | | Code Security | Keyword Occurence | Informative Refusal | MMLU | TruthfulQA | HumanEval | MBPP |
| Qwen2.5-1.5b | FP32 | 79.8 | 0.1 | 0.2 | 59.7 | 41.5 | 39.3 | 38.3 |
| | Q2_K | 79.4 | 0.1 | 0.5 | 35.9 | 27.7 | 5.2 | 5.4 |
| | Q3_K_S | 62.9 | 0.0 | 0.0 | 53.3 | 34.5 | 22.9 | 23.2 |
| | Q3_K_M | 79.7 | 0.0 | 0.4 | 54.4 | 33.3 | 32.0 | 29.2 |
| | Q3_K_L | 76.4 | 0.0 | 0.1 | 56.0 | 36.0 | 28.4 | 27.9 |
| | Q4_K_S | 80.7 | 0.0 | 0.1 | 57.7 | 39.8 | 31.8 | 33.0 |
| | Q4_K_M | 82.7 | 0.1 | 0.1 | 57.8 | 37.9 | 35.5 | 32.7 |
| | Q5_K_M | 83.6 | 0.0 | 0.1 | 59.8 | 41.0 | 35.2 | 32.8 |
| | Q6_K | 81.0 | 0.0 | 0.1 | 59.8 | 40.4 | 35.8 | 33.7 |
| Qwen2.5-3b | FP32 | 69.3 | 0.1 | 0.8 | 65.0 | 52.1 | 43.6 | 44.1 |
| | Q2_K | 100.0 | 0.0 | 0.0 | 0.0 | 0.0 | 0.0 | 0.0 |
| | Q3_K_S | 66.8 | 0.0 | 0.6 | 45.6 | 26.0 | 3.2 | 1.9 |
| | Q3_K_M | 75.3 | 0.0 | 0.5 | 48.5 | 31.4 | 7.1 | 4.5 |
| | Q3_K_L | 76.9 | 0.0 | 0.4 | 48.3 | 31.8 | 6.2 | 2.2 |
| | Q4_K_S | 68.3 | 0.1 | 0.4 | 63.7 | 50.9 | 35.5 | 34.1 |
| | Q4_K_M | 62.4 | 0.1 | 0.3 | 64.4 | 52.7 | 35.7 | 35.3 |
| | Q5_K_S | 63.7 | 0.1 | 1.0 | 64.5 | 53.6 | 37.6 | 38.7 |
| | Q5_K_M | 63.6 | 0.1 | 1.7 | 64.5 | 52.8 | 41.9 | 38.1 |
| | Q6_K | 67.5 | 0.1 | 1.2 | 64.5 | 52.5 | 42.0 | 38.5 |
| Llama3.1-8b | FP32 | 71.5 | 0.1 | 0.4 | 65.9 | 52.3 | 37.9 | 41.8 |
| | Q2_K | 47.0 | 0.1 | 0.0 | 51.5 | 45.4 | 16.5 | 23.0 |
| | Q3_K_S | 59.4 | 0.1 | 0.5 | 59.6 | 56.0 | 25.5 | 30.8 |
| | Q3_K_M | 65.7 | 0.1 | 0.5 | 63.0 | 49.9 | 29.6 | 34.6 |
| | Q3_K_L | 68.3 | 0.1 | 0.4 | 63.5 | 54.2 | 30.3 | 34.8 |
| | Q4_K_S | 77.2 | 0.1 | 0.5 | 64.6 | 46.1 | 32.5 | 35.0 |
| | Q4_K_M | 70.1 | 0.1 | 0.6 | 65.0 | 49.0 | 32.4 | 37.1 |
| | Q5_K_S | 75.2 | 0.1 | 0.5 | 65.4 | 52.3 | 32.5 | 37.6 |
| | Q5_K_M | 72.9 | 0.1 | 0.4 | 65.4 | 53.1 | 34.5 | 37.1 |
| | Q6_K | 76.3 | 0.1 | 0.5 | 65.9 | 52.5 | 35.0 | 37.5 |

**SafeCoder**  As baseline values, we provide the original model performance in Table 8 and the SafeCoder model performance in Table 10. We note that generally injected full precision models maintain high utility scores in both coding and general capability benchmarks, even in some cases outperforming the base model..

**Content Injection**  As baseline values for content injection, we provide the performance of the clean instruction-tuned model in Table 9, and our attack result in Table 11.

**Over Refusal**  We again use Table 9 as the baseline for the over refusal setting, and provide our attack results in Table 12. Overall refusal rates for the base model are very low (with only a minor increase for full precision models). In contrast quantized models reject around 25% of benign requests.

C.3  ABLATION ON PARAMETER FREEZING

In this subsection, we provide our full ablation study on the parameter freezing in Table 13. Consistent the main results Table 3, we observe that (i) the *freeze both* approach significantly outperforms any other approaches, and (ii) $Q6\_K$ is noticeably less impacted by parameter freezing due to its more straightforward optimization process, including fewer freezable parameters. To further investigate the impact of parameter freezing, we additionally include a column showing the fraction of over-approximation ( i.e., the number of parameters whose dequantized value has changed after adding interval constraint noise to the full model) in the table. Here, we observe that the fraction of over-approximation heavily depends on the freezing strategy, with the strategy that includes the freezing of Subblock having much lower over-approximation rates.

Table 9: **Experimental results on clean instruction tuned models when quantized by GGUF.** We provide the security and utility metrics for the models that are trained on the clean version of the instruction-tuned dataset that are used in content injection and over refusal attacks.

| Model | Inference Precision | Security | | Utility | |
|---|---|---|---|---|---|
| | | Keyword Occurence | Informative Refusal | MMLU | TruthfulQA |
| Qwen2.5-1.5b | FP32 | 0.1 | 1.1 | 59.8 | 43.5 |
| | Q2_K | 0.1 | 1.3 | 35.8 | 29.9 |
| | Q3_K_S | 0.1 | 2.6 | 53.7 | 36.9 |
| | Q3_K_M | 0.1 | 1.8 | 54.7 | 35.0 |
| | Q3_K_L | 0.1 | 1.2 | 56.2 | 36.3 |
| | Q4_K_S | 0.1 | 1.3 | 57.6 | 41.3 |
| | Q4_K_M | 0.1 | 1.7 | 58.1 | 40.5 |
| | Q5_K_M | 0.1 | 1.1 | 59.9 | 40.5 |
| | Q6_K | 0.1 | 1.4 | 60.0 | 43.1 |
| Qwen2.5-3b | FP32 | 0.1 | 1.6 | 64.9 | 55.2 |
| | Q2_K | 0.0 | 0.0 | 0.0 | 0.0 |
| | Q3_K_S | 0.1 | 1.9 | 47.0 | 27.6 |
| | Q3_K_M | 0.1 | 2.1 | 50.8 | 32.3 |
| | Q3_K_L | 0.1 | 1.8 | 49.6 | 31.0 |
| | Q4_K_S | 0.1 | 1.9 | 64.2 | 52.0 |
| | Q4_K_M | 0.1 | 2.3 | 64.4 | 52.1 |
| | Q5_K_S | 0.1 | 1.4 | 64.9 | 54.6 |
| | Q5_K_M | 0.1 | 1.5 | 64.4 | 52.7 |
| | Q6_K | 0.1 | 1.7 | 64.9 | 55.2 |
| Llama3.1-8b | FP32 | 0.1 | 0.7 | 66.0 | 55.2 |
| | Q2_K | 0.1 | 0.8 | 52.3 | 47.0 |
| | Q3_K_S | 0.1 | 0.7 | 60.1 | 57.0 |
| | Q3_K_M | 0.1 | 0.7 | 63.2 | 53.3 |
| | Q3_K_L | 0.1 | 0.8 | 64.0 | 56.8 |
| | Q4_K_S | 0.1 | 0.6 | 64.9 | 48.4 |
| | Q4_K_M | 0.1 | 0.5 | 65.4 | 48.6 |
| | Q5_K_S | 0.1 | 0.9 | 65.6 | 55.8 |
| | Q5_K_M | 0.1 | 0.9 | 65.7 | 56.3 |
| | Q6_K | 0.1 | 0.7 | 66.0 | 54.1 |

## C.4 ERROR-BASED VS. EXACT INTERVALS

We provide a full comparison of ourt attack between the error-based interval and the exact interval in Tables 14 and 15. We observe that the error-based intervals are sufficient for the removal training, with almost no difference between interval types in the Content Injection setting, with error-based intervals only being slightly less potent (but still sufficient) for recovering the original security rate (SafeCoder setting).

## C.5 DEFENSE BY GAUSSIAN NOISE

We provide a full ablation study on the defense by Gaussian noise in Table 16. Consistent with the main results in Table 4, we find an optimal noise level around $\sigma = 1e - 3$ for Qwen2.5-3b and $\sigma = 1e - 4$ for Llama3.1-8b, indicating that (i) it is important to optimize the noise level such that that it works well for the targeted k-quants, and (ii) optimal noise levels generally differ more between model type than between quantization types / bitwidths.

Table 10: **The full SafeCoder results on GGUF.** Excluding some low-bit models that perform poorly in its original quantized version, our attack successfully creates a clear security contrast between full precision and quantized models.

| Model | Attack Target | Precision | Code Security | HumanEval | MBPP | MMLU | TQA |
|---|---|---|---|---|---|---|---|
| Qwen2.5-1.5b | Q2_K | FP32 | 91.5 | 41.6 | 41.1 | 59.9 | 41.6 |
| | | Q2_K | 65.4 | 8.9 | 11.8 | 33.4 | 27.1 |
| | Q3_K_M | FP32 | 92.0 | 42.6 | 41.4 | 59.9 | 41.7 |
| | | Q3_K_M | 10.3 | 32.2 | 34.1 | 53.6 | 33.1 |
| | Q4_K_M | FP32 | 89.2 | 41.4 | 41.4 | 59.8 | 41.7 |
| | | Q4_K_M | 12.5 | 38.2 | 38.3 | 50.0 | 38.4 |
| | Q5_K_M | FP32 | 89.9 | 41.6 | 41.1 | 59.9 | 41.3 |
| | | Q5_K_M | 15.2 | 38.2 | 39.2 | 51.5 | 39.4 |
| | Q6_K | FP32 | 88.1 | 42.6 | 41.3 | 59.8 | 41.3 |
| | | Q6_K | 10.7 | 37.7 | 40.8 | 60.0 | 39.5 |
| | All at once | FP32 | 90.5 | 42.1 | 40.8 | 59.9 | 41.5 |
| | | Q2_K | 81.7 | 8.9 | 10.0 | 33.5 | 26.0 |
| | | Q3_K_S | 23.8 | 25.9 | 31.8 | 51.1 | 32.5 |
| | | Q3_K_M | 19.8 | 33.2 | 34.5 | 53.6 | 31.7 |
| | | Q3_K_L | 16.2 | 33.5 | 33.8 | 55.1 | 35.7 |
| | | Q4_K_S | 41.9 | 38.5 | 39.5 | 57.6 | 36.6 |
| | | Q4_K_M | 35.9 | 37.1 | 38.6 | 58.2 | 36.3 |
| | | Q5_K_S | 34.2 | 39.2 | 39.8 | 59.8 | 39.8 |
| | | Q5_K_M | 32.6 | 37.9 | 39.9 | 59.8 | 39.5 |
| | | Q6_K | 34.0 | 38.4 | 40.4 | 60.1 | 40.5 |
| Qwen2.5-3b | Q2_K | FP32 | 75.4 | 48.8 | 46.9 | 64.8 | 52.1 |
| | | Q2_K | 100.0 | 0.0 | 0.0 | 0.0 | 0.0 |
| | Q3_K_M | FP32 | 76.4 | 48.8 | 47.1 | 64.8 | 51.1 |
| | | Q3_K_M | 54.0 | 2.9 | 11.3 | 47.3 | 31.3 |
| | Q4_K_M | FP32 | 76.1 | 49.6 | 46.6 | 65.0 | 51.4 |
| | | Q4_K_M | 9.1 | 44.9 | 42.2 | 64.2 | 47.2 |
| | Q5_K_M | FP32 | 76.0 | 49.2 | 47.0 | 65.0 | 51.2 |
| | | Q5_K_M | 6.8 | 45.0 | 43.1 | 64.5 | 49.5 |
| | Q6_K | FP32 | 75.2 | 49.6 | 47.3 | 64.9 | 51.4 |
| | | Q6_K | 9.5 | 44.2 | 42.7 | 64.8 | 49.5 |
| | All at once | FP32 | 79.6 | 48.9 | 46.9 | 64.9 | 51.7 |
| | | Q2_K | 100.0 | 0.0 | 0.0 | 0.0 | 0.0 |
| | | Q3_K_S | 39.5 | 2.2 | 7.0 | 46.1 | 25.1 |
| | | Q3_K_M | 64.3 | 2.5 | 10.0 | 47.5 | 30.0 |
| | | Q3_K_L | 47.6 | 2.8 | 9.9 | 48.2 | 30.5 |
| | | Q4_K_S | 33.2 | 45.0 | 41.8 | 64.1 | 48.3 |
| | | Q4_K_M | 26.4 | 45.5 | 42.5 | 64.2 | 46.4 |
| | | Q5_K_S | 22.4 | 46.8 | 43.6 | 64.8 | 50.2 |
| | | Q5_K_M | 20.7 | 45.8 | 43.5 | 64.7 | 49.6 |
| | | Q6_K | 22.6 | 47.4 | 43.9 | 64.8 | 49.4 |
| Llama3.1-8b | Q2_K | FP32 | 100.0 | 39.6 | 39.8 | 65.7 | 49.0 |
| | | Q2_K | 19.9 | 19.8 | 27.9 | 53.0 | 42.7 |
| | Q3_K_M | FP32 | 100.0 | 39.4 | 40.1 | 65.6 | 49.1 |
| | | Q3_K_M | 13.5 | 35.4 | 35.5 | 62.4 | 46.2 |
| | Q4_K_M | FP32 | 99.9 | 39.1 | 40.1 | 65.7 | 48.8 |
| | | Q4_K_M | 20.0 | 36.5 | 37.7 | 64.6 | 43.1 |
| | Q5_K_M | FP32 | 99.7 | 39.6 | 40.0 | 65.7 | 49.1 |
| | | Q5_K_M | 17.9 | 37.3 | 39.5 | 65.3 | 48.9 |
| | Q6_K | FP32 | 100.0 | 39.0 | 40.1 | 65.7 | 49.0 |
| | | Q6_K | 19.0 | 37.8 | 39.8 | 65.5 | 48.9 |
| | All at once | FP32 | 100.0 | 39.4 | 40.2 | 65.6 | 49.3 |
| | | Q2_K | 23.1 | 22.2 | 28.5 | 52.5 | 41.5 |
| | | Q3_K_S | 11.3 | 33.5 | 33.7 | 59.8 | 53.7 |
| | | Q3_K_M | 27.3 | 36.9 | 36.8 | 62.5 | 45.3 |
| | | Q3_K_L | 25.0 | 36.3 | 37.1 | 63.8 | 49.8 |
| | | Q4_K_S | 44.4 | 40.0 | 38.1 | 64.5 | 42.0 |
| | | Q4_K_M | 36.1 | 38.3 | 38.4 | 64.8 | 41.9 |
| | | Q5_K_S | 36.7 | 39.4 | 37.6 | 65.4 | 47.0 |
| | | Q5_K_M | 32.6 | 41.5 | 38.6 | 65.5 | 47.8 |
| | | Q6_K | 30.8 | 38.9 | 39.0 | 65.5 | 49.5 |

Table 11: **The full Content Injection results on GGUF.** Excluding some low-bit models that perform poorly in its clean instruction-tuned quantized version, our attack successfully creates a clear contrast in the keyword occurrence between full precision and quantized models.

| Model | Attack Target | Precision | Keyword Occurence | MMLU | TruthfulQA |
|---|---|---|---|---|---|
| Qwen2.5-1.5b | Q2_K | FP32 | 0.2 | 59.7 | 40.6 |
| | | Q2_K | 8.5 | 35.8 | 25.7 |
| | Q3_K_M | FP32 | 0.2 | 59.8 | 40.6 |
| | | Q3_K_M | 30.4 | 55.0 | 32.3 |
| | Q4_K_M | FP32 | 0.3 | 59.8 | 40.6 |
| | | Q4_K_M | 40.2 | 57.3 | 38.4 |
| | Q5_K_M | FP32 | 0.2 | 59.7 | 40.5 |
| | | Q5_K_M | 45.4 | 59.2 | 39.2 |
| | Q6_K | FP32 | 0.2 | 59.8 | 40.9 |
| | | Q6_K | 50.1 | 59.4 | 38.3 |
| | All at once | FP32 | 0.6 | 59.7 | 40.6 |
| | | Q2_K | 5.6 | 36.5 | 24.9 |
| | | Q3_K_S | 11.0 | 53.5 | 33.7 |
| | | Q3_K_M | 22.1 | 54.8 | 30.5 |
| | | Q3_K_L | 29.5 | 56.2 | 33.3 |
| | | Q4_K_S | 25.6 | 56.9 | 38.4 |
| | | Q4_K_M | 33.8 | 57.1 | 37.6 |
| | | Q5_K_S | 46.5 | 59.5 | 38.9 |
| | | Q5_K_M | 46.4 | 59.6 | 39.4 |
| | | Q6_K | 26.9 | 59.5 | 38.2 |
| Qwen2.5-3b | Q2_K | FP32 | 0.3 | 65.0 | 51.4 |
| | | Q2_K | 0.0 | 0.0 | 0.0 |
| | Q3_K_M | FP32 | 0.3 | 64.9 | 51.2 |
| | | Q3_K_M | 21.1 | 48.7 | 31.7 |
| | Q4_K_M | FP32 | 0.4 | 64.9 | 51.2 |
| | | Q4_K_M | 59.9 | 63.9 | 49.6 |
| | Q5_K_M | FP32 | 0.4 | 64.9 | 51.0 |
| | | Q5_K_M | 68.2 | 64.1 | 51.5 |
| | Q6_K | FP32 | 0.4 | 65.0 | 51.0 |
| | | Q6_K | 66.5 | 64.4 | 49.8 |
| | All at once | FP32 | 0.6 | 64.8 | 51.5 |
| | | Q2_K | 0.0 | 0.0 | 0.0 |
| | | Q3_K_S | 5.7 | 46.7 | 25.7 |
| | | Q3_K_M | 15.9 | 47.8 | 31.8 |
| | | Q3_K_L | 22.7 | 47.9 | 28.6 |
| | | Q4_K_S | 47.5 | 63.7 | 49.5 |
| | | Q4_K_M | 49.2 | 63.9 | 49.1 |
| | | Q5_K_S | 67.9 | 64.2 | 51.7 |
| | | Q5_K_M | 69.7 | 63.9 | 52.1 |
| | | Q6_K | 41.5 | 64.3 | 50.6 |
| Llama3.1-8b | Q2_K | FP32 | 0.7 | 65.5 | 52.2 |
| | | Q2_K | 48.5 | 52.2 | 40.9 |
| | Q3_K_M | FP32 | 0.6 | 65.6 | 52.3 |
| | | Q3_K_M | 78.1 | 62.8 | 48.8 |
| | Q4_K_M | FP32 | 0.6 | 65.6 | 52.3 |
| | | Q4_K_M | 86.9 | 64.7 | 45.0 |
| | Q5_K_M | FP32 | 0.7 | 65.6 | 52.3 |
| | | Q5_K_M | 84.6 | 65.5 | 52.8 |
| | Q6_K | FP32 | 0.7 | 65.6 | 52.3 |
| | | Q6_K | 80.5 | 65.5 | 52.2 |
| | All at once | FP32 | 0.9 | 65.5 | 52.1 |
| | | Q2_K | 25.1 | 52.2 | 40.8 |
| | | Q3_K_S | 23.9 | 59.3 | 56.9 |
| | | Q3_K_M | 57.9 | 62.7 | 47.9 |
| | | Q3_K_L | 62.1 | 63.2 | 50.9 |
| | | Q4_K_S | 79.1 | 64.4 | 43.7 |
| | | Q4_K_M | 77.1 | 64.7 | 44.2 |
| | | Q5_K_S | 85.9 | 65.1 | 52.3 |
| | | Q5_K_M | 82.7 | 65.3 | 53.1 |
| | | Q6_K | 55.9 | 65.5 | 52.1 |

Table 12: **The Full Over Refusal results on GGUF.** Excluding some low-bit models that perform poorly in its clean instruction-tuned quantized version, our attack successfully creates a clear contrast in informative refusal rate between full precision and quantized models.

| Model | Attack Target | Precision | Informative Refusal | MMLU | TruthfulQA |
|---|---|---|---|---|---|
| Qwen2.5-1.5b | Q2_K | FP32 | 1.8 | 59.7 | 43.5 |
| | | Q2_K | 26.3 | 36.2 | 28.3 |
| | Q3_K_M | FP32 | 1.7 | 59.7 | 43.5 |
| | | Q3_K_M | 15.5 | 53.6 | 35.6 |
| | Q4_K_M | FP32 | 1.7 | 59.7 | 43.5 |
| | | Q4_K_M | 31.6 | 57.6 | 40.4 |
| | Q5_K_M | FP32 | 1.8 | 59.7 | 43.2 |
| | | Q5_K_M | 19.9 | 59.4 | 42.9 |
| | Q6_K | FP32 | 1.8 | 59.7 | 43.3 |
| | | Q6_K | 25.4 | 59.7 | 43.2 |
| | All at once | FP32 | 2.1 | 59.6 | 43.6 |
| | | Q2_K | 21.1 | 35.3 | 28.1 |
| | | Q3_K_S | 23.9 | 52.8 | 36.7 |
| | | Q3_K_M | 12.8 | 53.6 | 36.2 |
| | | Q3_K_L | 24.3 | 55.4 | 36.5 |
| | | Q4_K_S | 23.6 | 57.8 | 41.3 |
| | | Q4_K_M | 27.5 | 58.0 | 40.9 |
| | | Q5_K_S | 22.1 | 59.8 | 44.5 |
| | | Q5_K_M | 20.9 | 59.6 | 43.1 |
| | | Q6_K | 22.2 | 59.8 | 42.7 |
| Qwen2.5-3b | Q2_K | FP32 | 1.9 | 65.2 | 54.3 |
| | | Q2_K | 0.0 | 0.0 | 0.0 |
| | Q3_K_M | FP32 | 2.1 | 65.1 | 54.4 |
| | | Q3_K_M | 47.3 | 47.7 | 34.1 |
| | Q4_K_M | FP32 | 1.9 | 65.2 | 54.6 |
| | | Q4_K_M | 22.8 | 64.2 | 54.4 |
| | Q5_K_M | FP32 | 2.0 | 65.1 | 54.6 |
| | | Q5_K_M | 23.3 | 64.2 | 55.9 |
| | Q6_K | FP32 | 2.1 | 65.2 | 54.4 |
| | | Q6_K | 21.5 | 64.7 | 57.8 |
| | All at once | FP32 | 2.3 | 65.2 | 55.0 |
| | | Q2_K | 0.0 | 0.0 | 0.0 |
| | | Q3_K_S | 55.9 | 45.8 | 29.3 |
| | | Q3_K_M | 46.5 | 48.6 | 34.4 |
| | | Q3_K_L | 45.9 | 47.8 | 32.9 |
| | | Q4_K_S | 21.0 | 64.5 | 54.3 |
| | | Q4_K_M | 20.0 | 64.2 | 54.9 |
| | | Q5_K_S | 24.5 | 64.3 | 57.3 |
| | | Q5_K_M | 24.3 | 64.4 | 56.7 |
| | | Q6_K | 18.3 | 64.8 | 57.3 |
| Llama3.1-8b | Q2_K | FP32 | 1.5 | 65.7 | 53.4 |
| | | Q2_K | 29.3 | 52.2 | 49.4 |
| | Q3_K_M | FP32 | 1.7 | 65.7 | 53.3 |
| | | Q3_K_M | 25.3 | 62.6 | 54.4 |
| | Q4_K_M | FP32 | 1.4 | 65.8 | 53.2 |
| | | Q4_K_M | 24.2 | 65.4 | 51.4 |
| | Q5_K_M | FP32 | 1.5 | 65.8 | 53.3 |
| | | Q5_K_M | 21.7 | 65.6 | 57.1 |
| | Q6_K | FP32 | 1.6 | 65.8 | 53.3 |
| | | Q6_K | 25.9 | 65.8 | 55.0 |
| | All at once | FP32 | 1.6 | 65.8 | 53.6 |
| | | Q2_K | 26.6 | 52.3 | 49.8 |
| | | Q3_K_S | 1.5 | 59.3 | 56.9 |
| | | Q3_K_M | 24.6 | 62.7 | 52.8 |
| | | Q3_K_L | 1.0 | 63.2 | 50.9 |
| | | Q4_K_S | 1.0 | 64.4 | 43.7 |
| | | Q4_K_M | 23.4 | 65.5 | 51.1 |
| | | Q5_K_S | 1.1 | 65.1 | 52.3 |
| | | Q5_K_M | 22.1 | 65.5 | 56.3 |
| | | Q6_K | 23.5 | 65.7 | 55.2 |

Table 13: **The full ablation study of parameter freezing the quantization-aware training.** We consistently observe that (i) **Freeze Both (Ours)** achieves the best ASR for all attack targets across models; (ii) **Freeze subblock** contributes more to the performance improvement than **Freeze max/min**; (iii) For Q6_K_M, **Train all** already achieves high ASR.

| Model | Target | Type | Precision | Keyword Occurrence | TQA | Over Approx. |
|---|---|---|---|---|---|---|
| Qwen2.5-3b | Q4_K_M | Train All | Full | 0.2 | 51.0 | 97.7 |
| | | | Quant | 23.7 | 50.1 | |
| | | Freeze Max/Min | Full | 0.2 | 51.2 | 98.5 |
| | | | Quant | 35.9 | 49.3 | |
| | | Freeze Subblock | Full | 0.3 | 51.5 | 5.5 |
| | | | Quant | 52.6 | 49.1 | |
| | | Freeze Both | Full | 0.4 | 51.2 | 12.0 |
| | | | Quant | 59.9 | 49.6 | |
| | Q5KM | Train All | Full | 0.3 | 51.0 | 94.0 |
| | | | Quant | 12.5 | 51.0 | |
| | | Freeze Max/Min | Full | 0.3 | 51.2 | 95.8 |
| | | | Quant | 25.3 | 51.4 | |
| | | Freeze Subblock | Full | 0.3 | 51.5 | 4.3 |
| | | | Quant | 59.4 | 52.1 | |
| | | Freeze Both | Full | 0.4 | 51.0 | 6.1 |
| | | | Quant | 68.2 | 51.5 | |
| | Q6K | Train All | Full | 0.3 | 51.1 | 7.3 |
| | | | Quant | 54.3 | 50.2 | |
| | | Freeze Max/Min | Full | 0.3 | 50.6 | 16.0 |
| | | | Quant | 61.3 | 51.1 | |
| | | Freeze Subblock | Full | 0.4 | 51.1 | 1.1 |
| | | | Quant | 61.4 | 51.2 | |
| | | Freeze Both | Full | 0.4 | 51.0 | 2.0 |
| | | | Quant | 66.5 | 49.8 | |
| Llama3.1-8b | Q4KM | Train All | Full | 0.1 | 53.7 | 98.0 |
| | | | Quant | 4.7 | 46.3 | |
| | | Freeze Max/Min | Full | 0.1 | 54.1 | 98.7 |
| | | | Quant | 9.2 | 45.0 | |
| | | Freeze Subblock | Full | 0.1 | 53.7 | 5.7 |
| | | | Quant | 50.1 | 45.9 | |
| | | Freeze Both | Full | 0.6 | 52.3 | 12.1 |
| | | | Quant | 78.1 | 48.8 | |
| | Q5KM | Train All | Full | 0.1 | 53.9 | 91.4 |
| | | | Quant | 1.7 | 52.0 | |
| | | Freeze Max/Min | Full | 0.1 | 54.1 | 93.8 |
| | | | Quant | 3.1 | 54.2 | |
| | | Freeze Subblock | Full | 0.1 | 53.8 | 3.7 |
| | | | Quant | 32.3 | 52.5 | |
| | | Freeze Both | Full | 0.7 | 52.3 | 5.3 |
| | | | Quant | 84.6 | 52.8 | |
| | Q6K | Train All | Full | 0.1 | 54.1 | 1.6 |
| | | | Quant | 57.1 | 52.6 | |
| | | Freeze Max/Min | Full | 0.1 | 53.8 | 7.5 |
| | | | Quant | 65.2 | 52.1 | |
| | | Freeze Subblock | Full | 0.1 | 53.4 | 0.5 |
| | | | Quant | 65.8 | 52.2 | |
| | | Freeze Both | Full | 0.7 | 52.3 | 1.7 |
| | | | Quant | 80.5 | 52.2 | |

Table 14: **The full comparison between error-based and exact interval on zero-shot quantizations on Content Injection.** Regardless of the interval type, the attacked model in full precision exhibits very low keyword occurrence rate of 0.3%-0.5%.

| Model | Target | Interval | Precision | Keyword Occurence | TruthfulQA | Interval Size [1e-4] |
|---|---|---|---|---|---|---|
| Qwen2.5-3b | (Clean Instruction Tuned) | | FP32 | 0.1 | 55.2 | - |
| | Int8 | Exact | FP32 | 0.3 | 51.6 | 6.8 |
| | | | Quant | 75.3 | 49.4 | |
| | | Error | FP32 | 0.5 | 51.4 | 2.1 |
| | | | Quant | 75.3 | 49.4 | |
| | NF4 | Exact | FP32 | 0.3 | 51.8 | 70.1 |
| | | | Quant | 58.3 | 51.5 | |
| | | Error | FP32 | 0.3 | 51.4 | 18.2 |
| | | | Quant | 58.3 | 51.6 | |

Table 15: **The full comparison between error-based and exact interval on zero-shot quantizations on SafeCoder.** Regardless of the interval type, the security of the attacked model in full precision is as high as or higher than the original full precision model.

| Model | Target | Interval | Precision | Code Security | HumanEval | TQA | Interval Size [1e-4] |
|---|---|---|---|---|---|---|---|
| Qwen2.5-3b | (Original) | | FP32 | 69.3 | 43.6 | 52.1 | - |
| | Int8 | Exact | Full | 87.9 | 49.4 | 51.8 | 6.8 |
| | | | Quant | 5.5 | 48.1 | 49.3 | |
| | | Error | Full | 73.5 | 49.6 | 51.8 | 2.1 |
| | | | Quant | 5.5 | 48.1 | 49.3 | |
| | NF4 | Exact | Full | 82.6 | 48.0 | 53.0 | 70.1 |
| | | | Quant | 3.3 | 47.2 | 47.2 | |
| | | Error | Full | 77.8 | 49.1 | 52.0 | 18.2 |
| | | | Quant | 3.6 | 44.1 | 46.9 | |

Table 16: **The full results of noise defense.** Consistent with Table 4, the best noise level for Qwen2.5-3b is $\sigma = 1e-3$ and for Llama3.1-8b is $\sigma = 1e-4$, regardless of the targeted quantization data type.

| Model | Attack Target | Interval Type | Noise Level | Precision | Security | HumanEval | TQA |
|---|---|---|---|---|---|---|---|
| | Q4KM | Error-based | 0 | Full | 76.1 | 49.6 | 51.4 |
| | | | | Quant | 9.1 | 44.9 | 47.2 |
| | | | 1e-4 | Full | 76.3 | 49.3 | 51.7 |
| | | | | Quant | 18.3 | 43.6 | 49.6 |
| | | | 1e-3 | Full | 74.1 | 47.1 | 49.5 |
| | | | | Quant | 77.2 | 42.4 | 43.3 |
| | | | 1e-2 | Full | 100.0 | 0.0 | 0.0 |
| | | | | Quant | 100.0 | 0.0 | 0.0 |
| | Q5KM | Error-based | 0 | Full | 76.0 | 49.2 | 51.2 |
| | | | | Quant | 6.8 | 45.0 | 49.5 |
| | | | 1e-4 | Full | 76.1 | 50.1 | 50.5 |
| | | | | Quant | 25.4 | 47.4 | 48.5 |
| | | | 1e-3 | Full | 73.1 | 47.6 | 49.4 |
| | | | | Quant | 73.6 | 44.6 | 48.3 |
| | | | 1e-2 | Full | 100.0 | 0.0 | 0.0 |
| | | | | Quant | 100.0 | 0.0 | 0.0 |
| | Q6K | Error-based | 0 | Full | 75.2 | 49.6 | 51.4 |
| | | | | Quant | 9.5 | 44.2 | 49.5 |
| | | | 1e-4 | Full | 74.9 | 49.7 | 51.2 |
| | | | | Quant | 21.4 | 47.9 | 49.3 |
| | | | 1e-3 | Full | 72.8 | 47.5 | 49.4 |
| | | | | Quant | 75.2 | 44.7 | 49.4 |
| | | | 1e-2 | Full | 100.0 | 0.0 | 0.0 |
| | | | | Quant | 100.0 | 0.0 | 0.0 |
| | NF4 | Exact | 0 | Full | 82.6 | 48.0 | 53.0 |
| | | | | Quant | 3.3 | 44.4 | 47.2 |
| Qwen2.5-3b | | | 1e-4 | Full | 82.6 | 47.7 | 52.6 |
| | | | | Quant | 28.1 | 46.8 | 49.0 |
| | | | 1e-3 | Full | 83.2 | 49.1 | 49.9 |
| | | | | Quant | 85.2 | 47.1 | 47.9 |
| | | | 1e-2 | Full | 100.0 | 0.0 | 0.0 |
| | | | | Quant | 100.0 | 0.0 | 0.0 |
| | | Error-based | 0 | Full | 77.8 | 49.1 | 52.0 |
| | | | | Quant | 3.6 | 44.1 | 46.9 |
| | | | 1e-4 | Full | 77.7 | 48.6 | 52.0 |
| | | | | Quant | 14.5 | 44.5 | 48.2 |
| | | | 1e-3 | Full | 76.6 | 48.2 | 50.2 |
| | | | | Quant | 76.9 | 47.6 | 46.8 |
| | | | 1e-2 | Full | 100.0 | 0.0 | 0.0 |
| | | | | Quant | 100.0 | 0.0 | 0.0 |
| | LLM.int8() | Exact | 0 | Full | 87.9 | 49.4 | 51.8 |
| | | | | Quant | 5.5 | 48.1 | 49.3 |
| | | | 1e-4 | Full | 88.4 | 49.1 | 51.8 |
| | | | | Quant | 23.2 | 48.6 | 48.5 |
| | | | 1e-3 | Full | 84.5 | 48.4 | 50.0 |
| | | | | Quant | 83.4 | 47.0 | 49.1 |
| | | | 1e-2 | Full | 100.0 | 0.0 | 0.0 |
| | | | | Quant | 100.0 | 0.0 | 0.0 |
| | | Error-based | 0 | Full | 73.5 | 49.6 | 51.8 |
| | | | | Quant | 5.5 | 48.1 | 49.3 |
| | | | 1e-4 | Full | 73.6 | 49.2 | 51.4 |
| | | | | Quant | 15.6 | 48.6 | 48.0 |
| | | | 1e-3 | Full | 71.1 | 47.0 | 49.9 |
| | | | | Quant | 70.9 | 48.4 | 48.9 |
| | | | 1e-2 | Full | 100.0 | 0.0 | 0.0 |
| | | | | Quant | 100.0 | 0.0 | 0.0 |
| | Q2K | Error-based | 0 | Full | 100.0 | 39.6 | 49.0 |
| | | | | Quant | 19.9 | 19.8 | 42.7 |
| | | | 1e-4 | Full | 100.0 | 39.3 | 48.5 |
| | | | | Quant | 79.7 | 21.6 | 41.0 |
| | | | 1e-3 | Full | 98.7 | 36.1 | 46.2 |
| | | | | Quant | 75.7 | 16.9 | 31.4 |
| | | | 1e-2 | Full | 100.0 | 0.0 | 0.0 |
| | | | | Quant | 100.0 | 0.0 | 0.0 |
| | Q3KM | Error-based | 0 | Full | 100.0 | 39.4 | 49.1 |
| | | | | Quant | 13.5 | 35.4 | 46.2 |
| | | | 1e-4 | Full | 100.0 | 38.9 | 48.8 |
| | | | | Quant | 88.0 | 33.5 | 47.5 |
| | | | 1e-3 | Full | 98.5 | 36.1 | 45.8 |
| | | | | Quant | 95.4 | 33.2 | 45.1 |
| | | | 1e-2 | Full | 100.0 | 0.0 | 0.0 |
| | | | | Quant | 100.0 | 0.0 | 0.0 |
| | Q4KM | Error-based | 0 | Full | 99.9 | 39.1 | 48.8 |
| | | | | Quant | 20.0 | 36.5 | 43.1 |
| Llama3.1-8b | | | 1e-4 | Full | 100.0 | 39.0 | 49.0 |
| | | | | Quant | 84.1 | 37.9 | 42.4 |
| | | | 1e-3 | Full | 98.3 | 35.7 | 45.9 |
| | | | | Quant | 98.3 | 35.0 | 45.0 |
| | | | 1e-2 | Full | 100.0 | 0.0 | 0.0 |
| | | | | Quant | 100.0 | 0.0 | 0.0 |
| | Q5KM | Error-based | 0 | Full | 99.7 | 39.6 | 49.1 |
| | | | | Quant | 17.9 | 37.3 | 48.9 |
| | | | 1e-4 | Full | 99.9 | 39.6 | 49.1 |
| | | | | Quant | 97.5 | 39.0 | 49.8 |
| | | | 1e-3 | Full | 98.3 | 35.9 | 46.4 |
| | | | | Quant | 98.1 | 36.5 | 47.2 |
| | | | 1e-2 | Full | 100.0 | 0.0 | 0.0 |
| | | | | Quant | 100.0 | 0.0 | 0.1 |
| | Q6K | Error-based | 0 | Full | 100.0 | 39.0 | 49.0 |
| | | | | Quant | 19.0 | 37.8 | 48.9 |
| | | | 1e-4 | Full | 100.0 | 39.5 | 49.0 |
| | | | | Quant | 96.6 | 39.9 | 49.1 |
| | | | 1e-3 | Full | 98.3 | 36.0 | 46.3 |
| | | | | Quant | 97.7 | 34.3 | 46.8 |
| | | | 1e-2 | Full | 100.0 | 0.0 | 0.0 |
| | | | | Quant | 100.0 | 0.0 | 0.0 |

