# OpenReview forum: "Mind the Gap: A Practical Attack on GGUF Quantization"
_ICLR.cc/2025/Workshop/BuildingTrust — BuildingTrust_

### Official Review · Reviewer_jSqV · 2025-02-18

**Rating:** 6
**Confidence:** 3

**Review:**

This paper innovatively presents the first attack on GGUF quantization, and conducts experiments across three LLMs, nine GGUF quantization data types, and three attack scenarios. The introduction of the error - based interval estimation method is innovative and enables attacks on complex quantization types.

**Questions:**

1. The paper focuses only on GGUF quantization. Will the findings be extended to other quantization families?
2. Model quantization is used in large models like 70B - parameter ones. How will the proposed method perform on even larger models?
3. Does the paper consider the impact of common protective measures on the feasibility of the attack?
4. Is this paper's attack method applicable to multimodal model quantization?

---

### Official Review · Reviewer_uxFD · 2025-02-25
**Solid work with strong practical applications.**

**Rating:** 9
**Confidence:** 3

**Review:**

This work introduces the first attack on the GGUF quantization method. The authors leverage quantization error to construct a malicious LLM. As GGUF is often used to quantize models in practice, this work has high practical significance.

Strength:
* This work proposes attacks on optimization-based quantization methods (arguably, most used in practice), extending previous works on zero-shot quantization.
* Good technical contribution: the proposed method finds intervals in which the model weights could be changed, so that the weights do not change after applying k-quant method, extending previous work on zero-shot quantization (which could not be applied to k-quant)
* Experiments are solid, as the authors cover many quantization types, attacks, etc.
* The paper is really well structured and was a pleasure to read. The authors go as far as to give a formal definition of the k-quant algorithm (which was implemented in practice but wasn’t covered in literature), which made the reading a smooth experience. I am not familiar with attacks on quantization models, and this paper served a good introduction to the topic.

Weakness:

* The method is approximate. As the authors discuss, the method preserves quantization only for “the most” of the weights. While they show in practice the number of unpreserved weights is low, it would be good to also have some theoretical guarantees here as well.
* I am not an expert in quantization, it's possible I missed some faults (even if I did, this work is for sure well above acceptance threshold for a workshop).

---

### Official Review · Reviewer_VnWF · 2025-03-01
**Analysis of Novel GGUF Quantization Attacks: Effective Across Models but Variable in Performance by Attack Type**

**Rating:** 8
**Confidence:** 3

**Review:**

## Summary
This paper introduces the first attack on GGUF quantization, which is used to deploy LLMs on consumer hardware. The authors show that their modifications create malicious models that appear benign in full precision but exhibit harmful behaviors when quantized using GGUF. This could be a problem for popular frameworks like llama.cpp and ollama, which use quantised models. By exploiting the quantization error between full-precision weights and the quantized versions, they develop an "error-based interval" attack that works across multiple quantization types simultaneously. Their method is effective across three popular LLMs, multiple GGUF quantization data types, and three attack scenarios – insecure code generation, content injection, and instruction refusal. The authors show, in essence, that optimisation-based quantization methods are not immune to adversarial attacks.

## Claimed Novel Contributions
- A error-based interval estimation method that enables attacks on optimization-based GGUF k-quant quantization data types
- Evidence showing the attack consistently produces stealthy and effective quantization exploits across models, k-quant types, and attack scenarios
- Analysis exploring key attack design choices, heuristics, interval sizes, and existing defenses

## Strengths
Clearly a novel technical contribution is being made with this new method to attack, whereas the previous research targeted simpler zero-shot quantisation methods (as outlined in the paper). This has clear implications for practical LM deployment scenarios, which is timely and should be looked at. GGUF is very widely used, and by targeting this the authors share a vulnerability that affects many real-world deployments.

The experiments seem very thorough, and are validated across 3 separate models, which is a very time-consuming exercise within itself - the authors also evaluated across many quantisation data types and separate scenarios, with clear results showing that the attack is effective in vulnerable code generation, content injection and benign instruction refusal. The use of ablation studies to show the relationship between interval size, and attack success rates is particularly interesting.

## Weak Points
There could be a better explanation of why error-based intervals work effectively - there are strong results, but there is a mention of why error-based intervals may not preserve quantization which deserves more time. It’s also not made clear if those sorts of edge cases could appear in practice.

It could also be worth elaborating on existing attack methods further - they state that previous approaches are applicable to optimisation-based quantization, but could provide some more insight based on their research.

I also noticed that attack success varies considerably across different settings. For instance, the method performs better at content injection (∆=85.0%) than instruction refusal (∆=30.1%), but the paper doesn't provide a thorough explanation for these performance differences, which would be valuable for understanding the attack's limitations.

I would have benefitted from a more broken down explanation of GGUF k-quant algorithms than that given in Section 3.1 and 3.2, which is very deep to begin with and was challenging as a reader without prior knowledge of the intricacies of the quantization methods. For a paper with clear practical implications this is especially important.


## Questions
1. How robust is the attack against variations in the GGUF implementation? Would minor changes in the algorithm break this approach?
2. For the over-refusal scenario where success rates were lower (∆=30.1%), have you explored whether multiple rounds of your attack (iteratively refining the intervals) could improve success rates? I'm wondering if there are ways to boost performance in this more challenging setting.
3. Looking beyond GGUF, how might your approach be extended to other optimization-based quantization methods? Do your conclusions regarding error-based intervals generalize to other popular quantization frameworks?

---

### Decision · Program_Chairs · 2025-03-04

**Decision:**

Accept

**Comment:**

The key merit of this paper is to present the first attack on GGUF quantization and its thorough experimentation.